# SyncDiffusion: Coherent Montage via Synchronized Joint Diffusions

**Yuseung Lee**   **Kunho Kim**   **Hyunjin Kim**   **Minhyuk Sung**
KAIST
{phillip0701,kaist984,rlaguswls98,mhsung}@kaist.ac.kr

## Abstract

The remarkable capabilities of pretrained image diffusion models have been utilized not only for generating fixed-size images but also for creating panoramas. However, naive stitching of multiple images often results in visible seams. Recent techniques have attempted to address this issue by performing joint diffusions in multiple windows and averaging latent features in overlapping regions. However, these approaches, which focus on seamless montage generation, often yield incoherent outputs by blending different scenes within a single image. To overcome this limitation, we propose SYNCDIFFUSION, a plug-and-play module that synchronizes multiple diffusions through gradient descent from a perceptual similarity loss. Specifically, we compute the gradient of the perceptual loss using the predicted denoised images at each denoising step, providing meaningful guidance for achieving coherent montages. Our experimental results demonstrate that our method produces significantly more coherent outputs for text-guided panorama generation compared to previous methods (66.35% vs. 33.65% in our user study) while still maintaining fidelity (as assessed by GIQA) and compatibility with the input prompt (as measured by CLIP score). We further demonstrate the versatility of our method across three plug-and-play applications: layout-guided image generation, conditional image generation and 360-degree panorama generation. Our project page is at https://syncdiffusion.github.io.

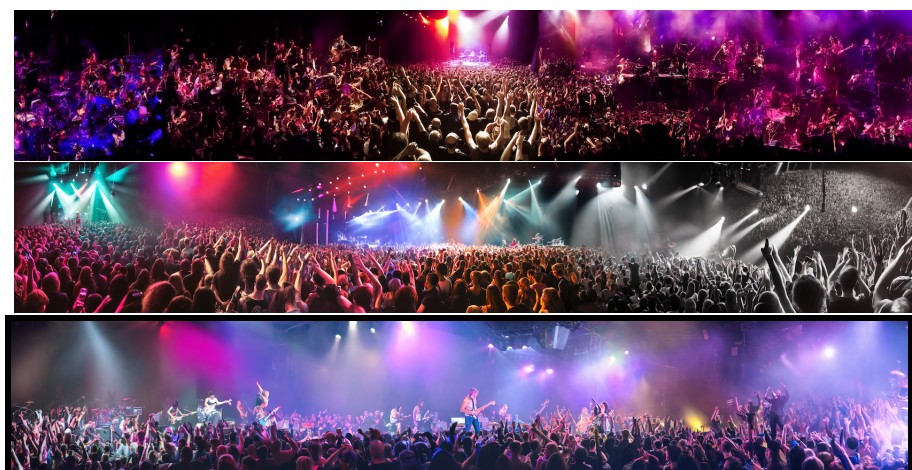

Figure 1: Comparison of panoramas generated with prompt *"A photo of a rock concert"* by Blended Latent Diffusion [1] (top), MultiDiffusion [3] (middle), and our SYNCDIFFUSION (bottom). Blended Latent Diffusion, when applied on image extrapolation, often generates visible seams and repetitive patterns. MultiDiffusion creates seamless panoramas but fails to achieve global coherence across the image. In contrast, our SYNCDIFFUSION synchronizes windows across the panorama by increasing the perceptual similarity of the denoised output predictions. This results in significantly more coherent panorama outputs.

37th Conference on Neural Information Processing Systems (NeurIPS 2023).

# 1 Introduction

Diffusion models have recently emerged as the forefront of generative models. Recent breakthroughs in text-to-image generation such as DALL·E 2 [36] and Stable Diffusion [39] are diffusion models trained with billions of images. Compared with GANs [16], diffusion models offer advantages not only in producing high-quality, realistic images but also in being utilized for conditional generation setups such as inpainting [30, 34, 41], editing [31, 18, 7, 1, 2, 34], and deblurring [48, 9], with few-shot [25, 12, 40] or even zero-shot [37, 36, 28, 46, 49] adaptation. The iterative reverse diffusion process can easily incorporate diverse conditions and regularizations at each step while guiding the entire process to produce realistic images. Hence, the diffusion model, once trained on a large-scale dataset, becomes a versatile and multi-purpose pretrained model that can be applied to various tasks and applications.

Recent work has extended the versatility of pretrained diffusion models to generate arbitrary-sized images or panoramas using either sequential [2] or joint [3] diffusion processes. Since typical image diffusion models are trained to generate fixed-sized images, creating panoramic images requires stitching multiple fixed-size images together, which can be impossible to do seamlessly without considering integration in the generation process. Two approaches have been proposed to tackle this issue. The first approach [2] involves generating the final output as a sequence of inpainting starting from an initial image, where each consecutive view image is produced while fixing the overlapped region (which is referred to as *image extrapolation* in their work [2]). However, this approach often struggles to seamlessly extend the given image and also tends to repeat similar patterns, resulting in unrealistic panoramas as shown in the first row of Fig. 1. The other approach is joint diffusion [3], which operates the reverse generative process simultaneously across multiple views while averaging the intermediate noisy images (or the noisy latent features) in the overlapped regions at each reverse process step. The blending of noisy latent features among the views at each denoising step can effectively generate a seamless montage of images. However, it is important to note that the content and styles of the images may vary across the views, resulting in a mixture of colorful and black-and-white images in a single panorama, as shown in the second row of Fig. 1. The lack of consistency occurs because the latent features of the overlapped regions are simply averaged without considering the coherence between them.

To address the limitation of previous work that produces unrealistic or incoherent montages, we present a novel synchronization module for joint diffusion, dubbed SYNCDIFFUSION. This module guides the reverse diffusion processes to achieve global coherence across different areas of the panorama image. Similar to previous guided diffusion methods [36, 30, 34, 41], our SYNCDIFFUSION guides the reverse diffusion process while adjusting the intermediate noisy images at each step. Our guidance is specifically provided as a gradient descent from a perceptual similarity loss calculated across multiple windows. Various off-the-shelf perceptual similarity losses such as LPIPS [53] or Style Loss [14] can be utilized in our framework. However, perceptual similarity losses computed with noisy images cannot effectively guide the denoising process. Thus, we draw inspiration from the non-Markovian formulation of DDIM [45] leveraging the prediction of the *denoised output* from the current noisy image at each denoising step. We compute the perceptual similarity loss using the *foreseen* denoised images at each step and then backpropagate the gradient through the noisy images. By leveraging the synergy with a prior seamless stitching technique [3] based on averaging latent features at each denoising step, our joint diffusion framework demonstrates the capability to generate montages that exhibit both local seamlessness and global coherence, as shown in the last row of Fig. 1. This is achieved in a zero-shot manner, without the need for retraining or fine-tuning of existing diffusion models.

In our experiments on text-guided panorama generation using Stable Diffusion 2.0 [39] model, the results demonstrate that our method achieves significantly higher coherence compared to previous methods. Quantitatively, as we increase the weight of the gradient descent, we observe improved coherence, measured by LPIPS [53] and Style Loss [14], while maintaining fidelity (measured by GIQA [17]) and compatibility with the input prompt (assessed by the CLIP score [19]). Diversity, measured by FID and KID, exhibits a trade-off with coherence, but our method still achieves much better scores compared to the baseline method. Our user studies confirm a significant preference for our method (66.35%) over the previous method (33.65%) in terms of coherence, while also suggesting superior image quality and higher prompt compatibility. Moreover, we further demonstrate the versabtility of SYNCDIFFUSION across three plug-and-play applications: layout-guided image generation, conditional image generation and 360-degree panorama generation.

## 2 Related Work

### 2.1 Diffusion Models

Diffusion probabilistic models [44, 11, 21, 33] are a group of generative models that generate data by sampling from an initial Gaussian distribution and iteratively applying a denoising process (referred to as the reverse process). These methods have achieved remarkable strides in image generation [39, 11, 42, 36], surpassing previous image generation models including GANs [24, 6]. DDPM [21] was among the pioneering models that showcased the impressive capability of image generation through Markovian forward and reverse processes, albeit with longer computation times in generation. This limitation was addressed by DDIM [45], which significantly reduced the sampling time in the reverse process using a non-Markovian transition formulation conditioned on the predicted denoised data. Furthermore, LDM [39] proposed incorporating the diffusion process into the latent space, achieving state-of-the-art realism in generated images and gaining attention in the text-to-image generation. Diffusion models have also demonstrated their applicability to diverse data modalities such as audio [50, 23, 29, 15], videos [22, 5], and 3D objects [35, 28, 46, 49].

### 2.2 Few-Shot or Zero-Shot Adaptation of Diffusion Models

Building upon the remarkable generation capabilities of pretrained public text-to-image diffusion models such as Stable Diffusion [39], recent research has introduced various methodologies for leveraging the pretrained models in diverse tasks including conditional generation, image editing, and manipulation, without the need to retrain the models from scratch. ControlNet [51] is an example of a method that enables the incorporation of additional conditions into existing text-to-image diffusion models through few-shot finetuning, wherein the image encoder is duplicated to handle the additional conditional image, and only a carefully selected subset of parameters is modified during the finetuning process. Custom Diffusion [25] also introduces a similar idea of enabling few-shot tuning while keeping the majority of parameters in the neural network frozen, but with applications of finetuning the model for a particular class or concept of images. Other previous work has also demonstrated that diffusion models can even be applied to novel tasks in a zero-shot manner. SDEdit [31] was the first to show zero-shot conditional image generation using a pretrained diffusion model by dispersing noise over the conditional image and denoising it back to a real image. RePaint [30] introduced an image inpainting idea by combining a generated foreground image and a noised background image at each time step. Similar *guided* diffusion ideas have also been explored for various tasks, such as image super-resolution [27, 13, 9, 43], colorization [41, 10], deblurring [48, 9], and style transfer [26, 25, 12, 40]. We propose a novel guided diffusion framework for image montage generation via joint diffusion.

### 2.3 Montage Generation via Diffusion Models

Panorama generation is one of the zero-shot applications of diffusion models. Since diffusion models are trained to generate images of a specific size and on a 2D plane, stitching is required to generate panoramas or textures. Most previous methods [38, 8, 2, 1] have employed inpainting-based approaches for seamless stitching. These methods extrapolate the accumulated image and fill only the missing regions to generate the panorama or texture. In contrast, MultiDiffusion [3] and DiffCollage [52] conduct diffusion in multiple views jointly while combining noisy latent features or scores at each reverse diffusion step. While both approaches have successfully produced continuous images, they have limitations in enforcing global coherence across the panorama or texture. MVDiffusion [47], a concurrent work, extends multi-view diffusion to produce non-square panorama images such as 360 panorama images by leveraging pixel-wise correspondence and attention modules However, it focuses on achieving smooth stitching, without addressing global coherence. In this work, we propose a simple yet effective synchronization module that can be integrated into any joint diffusion process to achieve global semantic coherence.

## 3 Backgrounds

### 3.1 Diffusion Models

In this section, we provide a brief overview of the Denoising Diffusion Probabilistic Models (DDPM) [21] and Denoising Diffusion Implicit Models (DDIM) [45], which are the foundations of recent pretrained image diffusion models. The aim of DDPM is to approximate the data distribution $q(\mathbf{x}_0)$ with a tractable model distribution $p_\theta(\mathbf{x}_0)$, which takes the form of a *Markov* chain with

learned Gaussian transitions $p_\theta(\mathbf{x}_{t-1}|\mathbf{x}_t)$ from $p(\mathbf{x}_T) = \mathcal{N}(\mathbf{x}_T; \mathbf{0}, \mathbf{I})$:

$$p_\theta(\mathbf{x}_0) = \int p_\theta(\mathbf{x}_{0:T}) d\mathbf{x}_{1:T}, \quad \text{where} \quad p_\theta(\mathbf{x}_{0:T}) = p(\mathbf{x}_T) \prod_{t=1}^{T} p_\theta(\mathbf{x}_{t-1}|\mathbf{x}_t). \tag{1}$$

The parameters of the joint distribution (known as the *reverse process*) $\theta$ are learned by minimizing the negative evidence lower bound (ELBO):

$$\min_\theta \mathbb{E}_{q(\mathbf{x}_0)} \left[ -\log p_\theta(\mathbf{x}_0) \right] \leq \min_\theta \mathbb{E}_{q(\mathbf{x}_0, \mathbf{x}_1, \cdots, \mathbf{x}_T)} \left[ -\log p_\theta(\mathbf{x}_{0:T}) + \log q(\mathbf{x}_{1:T}|\mathbf{x}_0) \right], \tag{2}$$

where $q_\theta(\mathbf{x}_{1:T}|\mathbf{x}_0)$ is the *forward process* adding a sequence of Gaussian noise to the data while increasing the noise scale. Among the variations of the forward processes, DDPM uses the *variance-preserving* diffusion that parameterizes the Gaussian transitions as follows with a decreasing sequence $\alpha_{1:T} \in (0, 1]^T$:

$$q(\mathbf{x}_{1:T}|\mathbf{x}_0) := \prod_{t=1}^{T} q(\mathbf{x}_t|\mathbf{x}_{t-1}), \text{ where } q(\mathbf{x}_t|\mathbf{x}_{t-1}) := \mathcal{N}\left( \sqrt{\frac{\alpha_t}{\alpha_{t-1}}} \mathbf{x}_{t-1}, \left( 1 - \frac{\alpha_t}{\alpha_{t-1}} \right) \mathbf{I} \right). \tag{3}$$

The definition of the Gaussian transitions in the forward process derives the following property:

$$q(\mathbf{x}_t|\mathbf{x}_0) := \mathcal{N}(\mathbf{x}_t; \sqrt{\alpha_t}\mathbf{x}_0, (1 - \alpha_t)\mathbf{I}), \tag{4}$$

and thus matches the choice of the starting distribution in the reverse process (a unit Gaussian) since $q(\mathbf{x}_T|\mathbf{x}_0)$ converges to a unit Gaussian when $\alpha_T$ is set close to 0. It also allows expressing $\mathbf{x}_t$ with $\mathbf{x}_0$ and a unit Gaussian noise variable $\epsilon$:

$$\mathbf{x}_t = \sqrt{\alpha_t}\mathbf{x}_0 + \sqrt{1 - \alpha_t}\epsilon, \quad \text{where} \quad \epsilon \sim \mathcal{N}(\mathbf{0}, \mathbf{I}). \tag{5}$$

In DDPM [21], the Gaussian transition $p_\theta(\mathbf{x}_{t-1}|\mathbf{x}_t)$ for each $\mathbf{x}_t$ in the reverse process is modeled as follows [1]:

$$p_\theta(\mathbf{x}_{t-1}|\mathbf{x}_t) := \mathcal{N}\left( \sqrt{\frac{\alpha_{t-1}}{\alpha_t}} \left( \mathbf{x}_t - \frac{1}{\sqrt{1-\alpha_t}} \left( 1 - \frac{\alpha_t}{\alpha_{t-1}} \right) \epsilon_\theta(\mathbf{x}_t, t) \right), \sigma_t^2 \mathbf{I} \right), \tag{6}$$

where $\sigma_t^2 = \frac{1-\alpha_{t-1}}{1-\alpha_t} \cdot \left( 1 - \frac{\alpha_t}{\alpha_{t-1}} \right)$, and $\epsilon_\theta(\mathbf{x}_t, t)$ is a learned function that optimizes the objective in Eq. 2 when it maps each $\mathbf{x}_t$ at time $t$ to a unit Gaussian noise, thus resulting in the following simplified loss:

$$L(\epsilon_\theta) := \sum_{t=1}^{T} \mathbb{E}_{\mathbf{x}_0 \sim q(\mathbf{x}_0), \epsilon_t \sim \mathcal{N}(\mathbf{0}, \mathbf{I})} \left[ \|\epsilon_\theta \left( \sqrt{\alpha_t}\mathbf{x}_0 + \sqrt{1 - \alpha_t}\epsilon_t, t \right) - \epsilon_t \|_2^2 \right]. \tag{7}$$

DDIM [45] provides a different perspective of seeing the same forward process as a *non-Markovian* process while taking the input data $\mathbf{x}_0$ into consideration in *reversed* transitions:

$$q(\mathbf{x}_{1:T}|\mathbf{x}_0) := q(\mathbf{x}_T|\mathbf{x}_0) \prod_{t=2}^{T} q(\mathbf{x}_{t-1}|\mathbf{x}_t, \mathbf{x}_0), \quad \text{where} \quad q(\mathbf{x}_T|\mathbf{x}_0) = \mathcal{N}(\sqrt{\alpha_T}\mathbf{x}_0, (1 - \alpha_T)\mathbf{I})$$

$$\text{and} \quad \forall s < t, \quad q(\mathbf{x}_s|\mathbf{x}_t, \mathbf{x}_0) = \mathcal{N}\left( \sqrt{\alpha_s}\mathbf{x}_0 + \sqrt{1 - \alpha_s - \sigma_t^2} \cdot \frac{\mathbf{x}_t - \sqrt{\alpha_t}\mathbf{x}_0}{\sqrt{1 - \alpha_t}}, \sigma_t^2 \mathbf{I} \right). \tag{8}$$

Then, each transition in the reverse process is also redefined as first predicting the *denoised observation* $\mathbf{x}_0$ given each $\mathbf{x}_t$ and then sampling $\mathbf{x}_{t-1}$ via the conditional distribution $q(\mathbf{x}_{t-1}|\mathbf{x}_t, \mathbf{x}_0)$:

$$p_\theta(\mathbf{x}_{t-1}|\mathbf{x}_t) := \begin{cases} q(\mathbf{x}_{t-1}|\mathbf{x}_t, \phi_\theta(\mathbf{x}_t, t)) & \text{if } t \geq 2 \\ \mathcal{N}(\phi_\theta(\mathbf{x}_t, t), \sigma_t^2 \mathbf{I}) & \text{if } t = 1, \end{cases} \tag{9}$$

---

[1]Note that $\beta_t$ in the DDPM [21] is equivalent to $1 - \frac{\alpha_t}{\alpha_{t-1}}$ in our paper and DDIM [45].

where

$$\phi_\theta(\mathbf{x}_t, t) = \frac{1}{\sqrt{\alpha_t}}(\mathbf{x}_t - \sqrt{1 - \alpha_t}\epsilon_\theta(\mathbf{x}_t, t)) \tag{10}$$

is the predicted denoised observation. The key observations of DDIM are twofold. First, the same simplified objective (Eq. 7) can be used to find the best models $\epsilon_\theta(\mathbf{x}_t, t)$ in Eq. 10 that minimize the negative ELBO (Eq. 2). This means that the DDIM reverse process can be used with a pretrained DDPM without retraining. Second, a subset of the time sequence $[1, \cdots, T]$ can be used in the reverse process of DDIM since $\mathbf{x}_s$ for any $s < t$ can be sampled from $\mathbf{x}_t$ via the $\mathbf{x}_0$ prediction, enabling a significant boost in the reverse process computation.

In the rest of the paper, the operation sampling the next denoised data in the reverse process with the learned distribution $p_\theta(\mathbf{x}_{t-1}|\mathbf{x}_t)$ defined in either Eq. 6 (DDPM) or Eq. 9 (DDIM) is denoted as:

$$\mathcal{S}(\mathbf{x}_t, t, \epsilon), \tag{11}$$

which takes a noisy data $\mathbf{x}_t$ at timestep $t$ and a unit Gaussian noise $\epsilon \sim \mathcal{N}(\mathbf{0}, \mathbf{I})$ as input.

## 3.2  Joint Diffusion

In an image diffusion model, each sample from the data distribution is either a 2D grid of per-pixel colors, or a 2D grid of latent features (as in Latent Diffusion [39]) that can be encoded from or decoded to a real image through a pretrained encoder $\mathcal{E}$ and decoder $\mathcal{D}$. In the rest of the paper, the term *image* will thus be used to refer to either a *color* image or a *latent feature* image, unless explicitly stated otherwise. Image diffusion models pretrained on fixed-size images cannot be used directly to produce arbitrary-size images. MultiDiffusion [3] has addressed this limitation by using a multi-window joint diffusion approach. The framework integrates images generated from multiple windows seamlessly by *averaging* colors or features across the windows at *every* reverse diffusion step. For instance, consider the case of generating a panorama image $\mathbf{z} \in \mathbb{R}^{H_z \times W_z \times D}$. The image at each window $\mathbf{x}^{(i)} \in \mathbb{R}^{H_x \times W_x \times D}$ is a subarea of the panorama image whose union across all the windows covers the entire panorama image. Let $\mathbf{m}^{(i)} \in [0, 1]^{H_x \times H_x}$ denote a binary mask for the subregion in the panorama image corresponding to the $i$-th window. The function $\mathcal{T}_{\mathbf{z} \to i} : \mathbb{R}^{H_z \times W_z \times D} \to \mathbb{R}^{H_x \times W_x \times D}$ maps (crops) the panorama image $z$ to the $i$-th window image, while $\mathcal{T}_{i \to \mathbf{z}} : \mathbb{R}^{H_x \times W_x \times D} \to \mathbb{R}^{H_z \times W_z \times D}$ is its inverse function that fills the region outside of the mask $\mathbf{m}_i$ with zeros. During the joint diffusion process running the reverse process simultaneously for each window, the noisy images from the windows $\mathbf{x}_t^{(i)}$ are first averaged in the panorama space:

$$\mathbf{z}_t = \frac{\sum_i \mathcal{T}_{i \to \mathbf{z}}(\mathbf{x}_t^{(i)})}{\sum_i \mathbf{m}^{(i)}}, \tag{12}$$

and then, the resulting combined noisy image $\mathbf{z}_t$ is cropped again for each window $\mathbf{x}_t^{(i)} = \mathcal{T}_{\mathbf{z} \to i}(\mathbf{z}_t)$, modifying the noisy image at each window with a inter-window regularization.

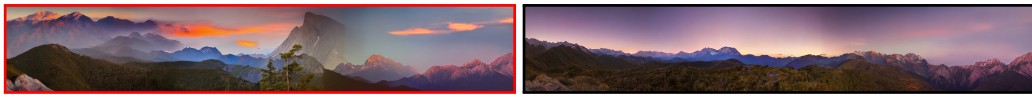

Figure 2:  Panoramas generated by MultiDiffusion [3] (left) and our SYNCDIFFUSION (right), with a prompt *"A photo of a mountain range at twilight"*. MultiDiffusion often combines various scenes, such as mountains with trees and snow, and even awkwardly blends them. In contrast, SYNCDIFFUSION generates panoramas that are significantly more coherent.

## 4  SyncDiffusion

While MultiDiffusion [3] can generate seamless panorama images from joint diffusion, it often fails to produce coherent and realistic montages. The left image in Fig. 2 demonstrates that the resulting image often oddly combines various scenes, such as mountains with trees and snow. Also, the blending occasionally fails to merge them in a realistic manner, as shown in the figure where distant objects are connected to closer objects. This incoherence issue in MultiDiffusion arises due to two main reasons. Firstly, the averaging operation only aligns the colors or features in the overlapped regions but does not match the *content* or *style* of the images. Secondly, it only enforces *adjacent* views to influence each other, and thus global coherence between distant windows cannot be achieved.

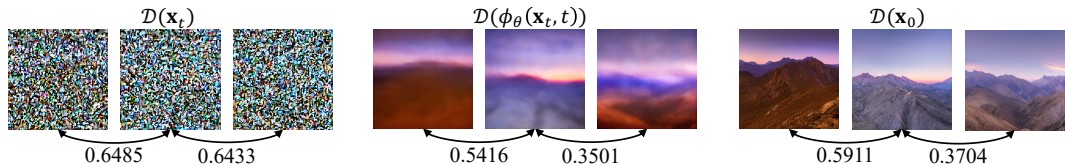

$\mathcal{D}(\mathbf{x}_t)$              $\mathcal{D}(\phi_\theta(\mathbf{x}_t, t))$              $\mathcal{D}(\mathbf{x}_0)$

    0.6485    0.6433          0.5416    0.3501          0.5911    0.3704

Figure 3: LPIPS [53] scores computed across the noisy images $\mathcal{D}(\mathbf{x}_t)$ at the intermediate step ($t = 45$ out of 50) of the reverse process (left), the *predicted* denoised images $\mathcal{D}(\phi(\mathbf{x}_t, t))$ at the same timestep $t$ (middle), and the final generated images $\mathcal{D}(\mathbf{x}_0)$ at timestep $t = 0$ (right). The indistinguishable noisy images yield similar LPIPS scores among them, whereas the predicted denoised images, which closely resemble the final outputs even at the beginning of the denoising process, exhibit LPIPS scores that align with those of the final generated images. This indicates that the predicted denoised images can provide meaningful guidance for producing coherent panoramas in the diffusion process.

---

**Algorithm 1:** Pseudocode of one-time denoising in SYNCDIFFUSION.

---

**Parameters** : $w$;                                         `// Gradient descent weight.`

**Inputs:** $\{\mathbf{x}_t^{(i)}\}_{i=0\cdots N-1}$;                        `// Noisy images at timestep t.`

**Outputs:** $\{\mathbf{x}_{t-1}^{(i)}\}_{i=0\cdots N-1}$;                    `// Noisy images at timestep t − 1.`

1   **Function** SyncDiffusion($\{\mathbf{x}_t^{(i)}\}$):

2     $\hat{\mathbf{x}}_t^{(i)} \leftarrow \mathbf{x}_t^{(i)}$;                   `// The anchor window at index 0 is not changed.`

3     **for** $i = 1, \ldots, N-1$ **do**

4        $\hat{\mathbf{x}}_t^{(i)} \leftarrow \mathbf{x}_t^{(i)} - w \nabla_{\mathbf{x}_t^{(i)}} \mathcal{L}\left(\mathcal{D}(\phi_\theta(\mathbf{x}_t^{(i)}, t)), \mathcal{D}(\phi_\theta(\mathbf{x}_t^{(0)}, t))\right)$; `// Gradient descent (Eq. 14)`

5     **return** $\{\hat{\mathbf{x}}_t^{(i)}\}$;

6   **Function** MultiDiffusion($\{\tilde{\mathbf{x}}_t^{(i)}\}$):

7     $\mathbf{z}_t \leftarrow \frac{\sum_i \mathcal{T}_{i \to \mathbf{z}}(\tilde{\mathbf{x}}_t^{(i)})}{\sum_i \mathbf{m}^{(i)}}$;           `// Averaging in the global space (Eq. 12).`

8     **for** $i = 0, \ldots, N-1$ **do**

9        $\mathbf{x}_t^{(i)} \leftarrow \mathcal{T}_{\mathbf{z} \to i}(\mathbf{z}_t)$;

10     **return** $\{\mathbf{x}_t^{(i)}\}$;

11   **Function** DenoisingOneStep($\{\mathbf{x}_t^{(i)}\}$):

12     $\{\hat{\mathbf{x}}_t^{(i)}\} \leftarrow$ SyncDiffusion($\{\mathbf{x}_t^{(i)}\}$);

13     **for** $i = 0, \ldots, N-1$ **do**

14        $\tilde{\mathbf{x}}_{t-1}^{(i)} \leftarrow \mathcal{S}(\hat{\mathbf{x}}_t^{(i)}, t, \epsilon)$;           `// Sampling the next denoised data (Eq. 11).`

15     $\{\mathbf{x}_{t-1}^{(i)}\} \leftarrow$ MultiDiffusion($\{\tilde{\mathbf{x}}_{t-1}^{(i)}\}$);

16     **return** $\{\mathbf{x}_{t-1}^{(i)}\}$;

---

To address this problem, we introduce a module called SYNCDIFFUSION which enables the generation of coherent montages, as shown on the right in Fig. 2. This module can be easily integrated into an existing joint diffusion framework. Similar to MultiDiffusion, our SYNCDIFFUSION module updates the noisy image at every step of the reverse diffusion process. In contrast to averaging the colors or latent features in the overlapped regions, however, SYNCDIFFUSION employs the backpropagation of gradients from a perceptual similarity loss computed across the windows to perform the update. The perceptual similarity loss, denoted as $\mathcal{L}$, can utilize any off-the-shelf loss function for perceptual similarity, such as LPIPS [53] and Style Loss [14]. To facilitate efficient computation, we designate an *anchor* window with an index of 0. For each view's noisy color image $\mathcal{D}(\mathbf{x}_t^{(i)})$ and the anchor window's noisy color image $\mathcal{D}(\mathbf{x}_t^{(0)})$ (where the decoder $\mathcal{D}$ can be treated as an identity function if the given diffusion model operates in image space rather than latent space), one can measure the coherence using the images and conduct gradient descent through $\mathbf{x}_t^{(i)}$:

$$\hat{\mathbf{x}}_t^{(i)} = \mathbf{x}_t^{(i)} - w \nabla_{\mathbf{x}_t^{(i)}} \mathcal{L}\left(\mathcal{D}(\mathbf{x}_t^{(i)}), \mathcal{D}(\mathbf{x}_t^{(0)})\right), \tag{13}$$

where $w$ is the weight of the gradient descent. However, the coherence measured with the *noisy* images cannot provide meaningful guidance. Fig. 3 shows examples where the left three images are the intermediate noisy images $\mathcal{D}(\mathbf{x}_t)$ at timestep $t = 45$ out of a total of 50 timesteps in the DDIM

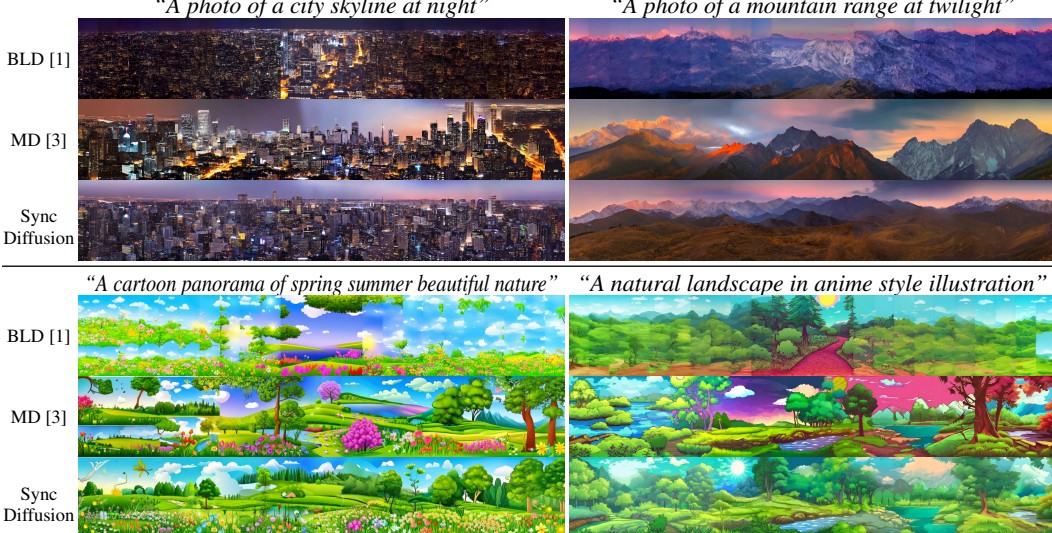

Figure 4: Qualitative comparisons. Blended Latent Diffusion [1] (BLD, the first row of each case) tends to exhibit visible seams and repetitive patterns. MultiDiffusion [3] (MD, the second row of each case) generates seamless results but lacks coherence, such as blending a sunset sky with a blue sky (top right), and displaying a combination of purple, pink, and blue backgrounds (bottom). In contrast, our SYNCDIFFUSION (the last in each case) produces seamless panoramas with significantly improved coherence. Best view in zoom and color.

reverse process. Note that the LPIPS scores among the noisy images are indistinguishable. Hence, similar to the DDIM reverse process, we utilize the *foreseen denoised* observation of each noisy data $\phi_\theta(\mathbf{x}_t^{(i)}, t)$ in Eq. 10. We measure the coherence not with the current noisy color images $\{\mathcal{D}(\mathbf{x}_t^{(i)})\}$ but with the predicted denoised color images $\{\mathcal{D}(\phi_\theta(\mathbf{x}_t^{(i)}, t))\}$ with the timestep $t$ and perform the backpropagation of the gradient through $\mathbf{x}_t^{(i)}$, resulting in the following updated formulation:

$$\hat{\mathbf{x}}_t^{(i)} = \mathbf{x}_t^{(i)} - w\nabla_{\mathbf{x}_t^{(i)}}\mathcal{L}\left(\mathcal{D}(\phi_\theta(\mathbf{x}_t^{(i)}, t)), \mathcal{D}(\phi_\theta(\mathbf{x}_t^{(0)}, t))\right). \tag{14}$$

In Fig. 3, the middle three images depict the predicted denoised images $\mathcal{D}(\phi_\theta(\mathbf{x}_t, t))$ at timestep $t = 45$, which closely resemble the final generated images $\mathcal{D}(\mathbf{x}_0)$ at timestep $t = 0$, even during the initial stages of the reverse diffusion process. Therefore, the LPIPS scores among the predicted denoised images also match those of the generated images, providing meaningful guidance for maintaining coherence. During each denoising step in the joint reverse process, we apply this update to the noisy images for all windows $\{\mathbf{x}_t^{(i)}\}$, and sample the one-step denoised images. MultiDiffusion is also applied to average the sampled images at the end. Refer to Alg. 1 for detailed pseudocode.

## 5 Results

### 5.1 Text-Guided Panorama Generation

In our experiments, we generate panorama images using our SYNCDIFFUSION method and the pretrained Stable Diffusion 2.0 [39] model. Stable Diffusion model operates in a latent space of $\mathbb{R}^{64\times64\times4}$ and generates images of $\mathbb{R}^{512\times512\times3}$. We generate panorama images of resolution $512 \times 3072$ ($64 \times 384$ in the latent space), where the width is six times the width of the output of Stable Diffusion. Each window $\mathbf{x}^{(i)}$ has an image resolution of $512 \times 512$, with a stride of 128 pixels along the width in the image space which is equivalent to stride 16 in the latent space, resulting in a total 21 windows to operate diffusion processes jointly. We use six text prompts from MultiDiffusion [3] (see S.8 of the **supplementary**) and generate 500 panoramas per prompt. For the gradient descent weight $w$ (Eq. 14), we experiment with various initial values while applying a weight decay with a rate of 0.95. We also set the center window as the anchor window with an index of 0.

**Baselines** We compare our SYNCDIFFUSION with previous methods that generate panoramas using a pretrained diffusion model. Blended Latent Diffusion [1] is an inpainting-based method that

extrapolates a single window image. MultiDiffusion [3], served as the base of our framework, is a special case of our method when the weight of gradient descent $w$ is 0. The same Stable Diffusion 2.0 model is used for all the methods for a fair comparison.

**Evaluation Metrics** We utilize a range of metrics to assess the coherence, fidelity, diversity, and compatibility of the output panoramas with the input prompt.

- (Coherence) **Intra-LPIPS** [53] and **Intra-Style-L** [14]: To assess the coherence of the generated panoramas, we introduce two metrics. Intra-LPIPS and Intra-Style-L, which are computed as the averages of LPIPS [53] and Style Loss [14], respectively, between a pair of non-overlapping window images from the same panorama. Specifically, we divide the panorama into 6 windows, each with dimensions of $512 \times 512$ and then compute the average of LPIPS and Style Loss across the 15 combinations of these cropped views. To provide a reference for the scale of these values, we generate 500 single-window-size images using the same Stable Diffusion model and compute the LPIPS and Style Loss for randomly selected 1,000 pairs of these reference images.
- (Fidelity) **Mean-GIQA** [17]: GIQA quantifies the fidelity of *individual* images by calculating the inverse of the distance between a query image and a reference set in a feature space. Mean-GIQA is computed by taking a *single* random crop of each panorama in $512 \times 512$ size and computing the average GIQA score from each cropped image to the reference set of images mentioned above.
- (Fidelity & Diversity) **FID** [20] and **KID** [4]: FID [20] and KID [4] are used to measure both fidelity and diversity. Both of them are measured with the aforementioned randomly cropped images and the set of reference images.
- (Compatibility with the Input Prompt) **Mean-CLIP-S** [19]: The compatibility with the input prompt is assessed using the mean of CLIP scores [19], denoted as Mean-CLIP-S. This metric is calculated using the same set of cropped images and the input prompt.

**Qualitative Comparisons** Fig. 4 showcases qualitative comparisons between our method and the baseline methods. Here, we show the results of our method generated with a weight parameter of $w = 20$. Blended Latent Diffusion [1] often exhibits visible seams due to the sequential inpainting scheme and produces repetitive patterns in the extrapolation, as illustrated by the mountains in the second case of the top row and the flowers and trees in both cases of the bottom row. MultiDiffusion [3] achieves seamless outputs, although it often produces incoherent outputs, such as mixing a sunset sky with a blue sky, as shown in the second cases of the first and second rows, and pink and purple backgrounds with blue backgrounds in the two cases of the bottom row. Our SYNCDIFFUSION generates visually and semantically more coherent panoramas with all the prompts. More qualitative comparisons are provided in the **supplementary (S.1, S.8)**.

**Quantitative Results** Fig. 5.1 presents a quantitative comparison among the methods. Our method's results are displayed using two different gradient descent weights, $w = 10$ and 20. MultiDiffusion [3] is also the case when $w = 0$ in our framework. For results obtained with different weights, please refer to the **supplementary (S.2)**. The color bars in the plots indicate the average scores across the six prompts, while the black lines depict the standard deviation. Note that as the gradient weight increases, both Intra-LPIPS and Intra-Style-L decrease. When $w = 20$, the Intra-LPIPS and Intra-Style-L of our method are approximately 3/4 and 1/6 of those computed with the reference set images (referred to as SD, Stable Diffusion), respectively, indicating significantly higher coherence. Moreover, the Mean-CLIP-S and Mean-GIQA scores are comparable to those computed with the reference set, meaning that the compatibility with the input prompt and fidelity are not compromised by our diffusion synchronization. The results of FID and KID demonstrate the trade-off between coherence and diversity. As the gradient descent weight $w$ increases, FID and KID also increase slightly, although they are still much lower compared to Blended Latent Diffusion [1]. This implies that for certain images, it is more difficult to find coherent images. In the **supplementary (S.3)**, we substantiate this claim with the results of shorter generated panoramas. Blended Latent Diffusion results in low Intra-LPIPS due to its tendency to repeat similar patterns, but it leads to low Mean-GIQA and very high FID and KID, indicating a significant degradation in fidelity.

**User Study** We conducted three user studies to further evaluate the coherence, image quality and prompt compatibility of the generated panoramas, respectively. Following Ritchie [32], participants were presented with panorama images generated by both MultiDiffusion [3] and our SYNCDIF-FUSION methods (with $w = 20$). They were then asked to choose one of them by answering the question: Which one appears a more coherent panorama image to you? **(Coherence)**, Which one is of higher quality? **(Image Quality)**, or Which one best matches the shared caption? **(Prompt Compatibility)**. We collected 25 responses each, including 5 vigilance

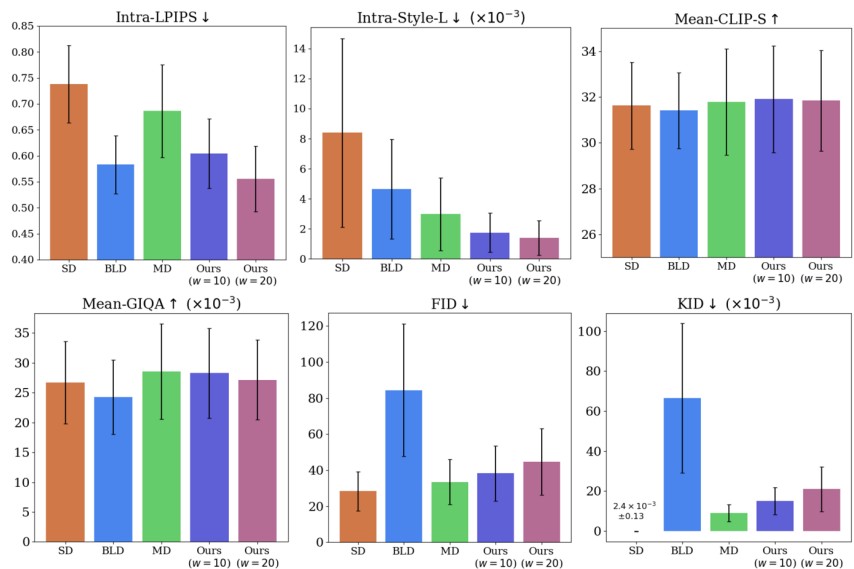

Figure 5: Quantitative results. MultiDiffusion [3] (MD) can be considered as a special case of our method when the gradient descent weight $w$ is set to 0. As $w$ increases, coherence (Intra-LPIPS and Intra-Style-L) improves while maintaining the compatibility with the input prompt (Mean-CLIP-S) and fidelity (Mean-GIQA). There is a trade-off between coherency and diversity, as indicated by the FID and KID results. Note that the FID and KID of our method are still significantly lower than those of Blended Latent Diffusion [1] (BLD). SD (Stable Diffusion) is the score with the reference set images. Refer to the text for the details.

tasks, from 100 participants for each user study. The results in Tab. 1 affirm that human evaluators perceive SYNCDIFFUSION as producing more coherent results compared to MultiDiffusion, while also demonstrating superior image quality and higher prompt compatibility. Refer to the **supplementary (S.7)** for detailed setups for the user study.

|  | Coherence (%) | Image Quality (%) | Prompt Compatibility (%) |
|---|---|---|---|
| MultiDiffusion [3] | 33.65 | 42.81 | 40.50 |
| SYNCDIFFUSION | **66.35** | **57.19** | **59.50** |

Table 1: User study results.

## 5.2 Additional Applications of SYNCDIFFUSION

We further demonstrate the versatility of SYNCDIFFUSION through three additional plug-and-play applications: layout-guided image generation, conditional image generation and 360-degree panorama generation.

**Layout-Guided Image Generation** Plugging SYNCDIFFUSION into the layout-to-image pipeline in MultiDiffusion [3] leads to a notable enhancement in the global coherence as displayed in Fig. 6-(A). While MultiDiffusion (middle row) generates an unnatural image with incoherent background around the house and the bear, our method produces a natural image with a globally coherent background.

**Conditional Image Generation** When integrated with ControlNet [51], SYNCDIFFUSION extends the conditional image generation to arbitrary resolutions. Let $c \in \mathbb{R}^{H_z \times W_z \times 3}$ denote an input condition and $\psi_{c \to i} : \mathbb{R}^{H_z \times W_z \times 3} \to \mathbb{R}^{H_x \times W_x \times 3}$ be a mapping from $c$ to the $i$-th cropped condition corresponding to the window $\mathbf{x}_t^{(i)}$. We define conditional SYNCDIFFUSION by substituting $\phi_\theta(\mathbf{x}_t, t)$ in Alg. 1 with $\phi_\theta(\mathbf{x}_t, t, c^{(i)})$, where $c^{(i)} := \psi_{c \to i}(c)$. Fig. 6-(B) illustrates that the combination of ControlNet and SYNCDIFFUSION generates coherent panoramas while reflecting the given condition Canny edge map (top row).

**360-degree Panorama Generation** We further plug SYNCDIFFUSION into MVDiffusion [47], a concurrent work that generates 360-degree panoramas from text prompts via multi-view diffusion. As shown Fig. 6-(C), our SYNCDIFFUSION distinctly improves the global coherence of the generated panorama. The increase in coherence becomes more apparent when comparing perspective views

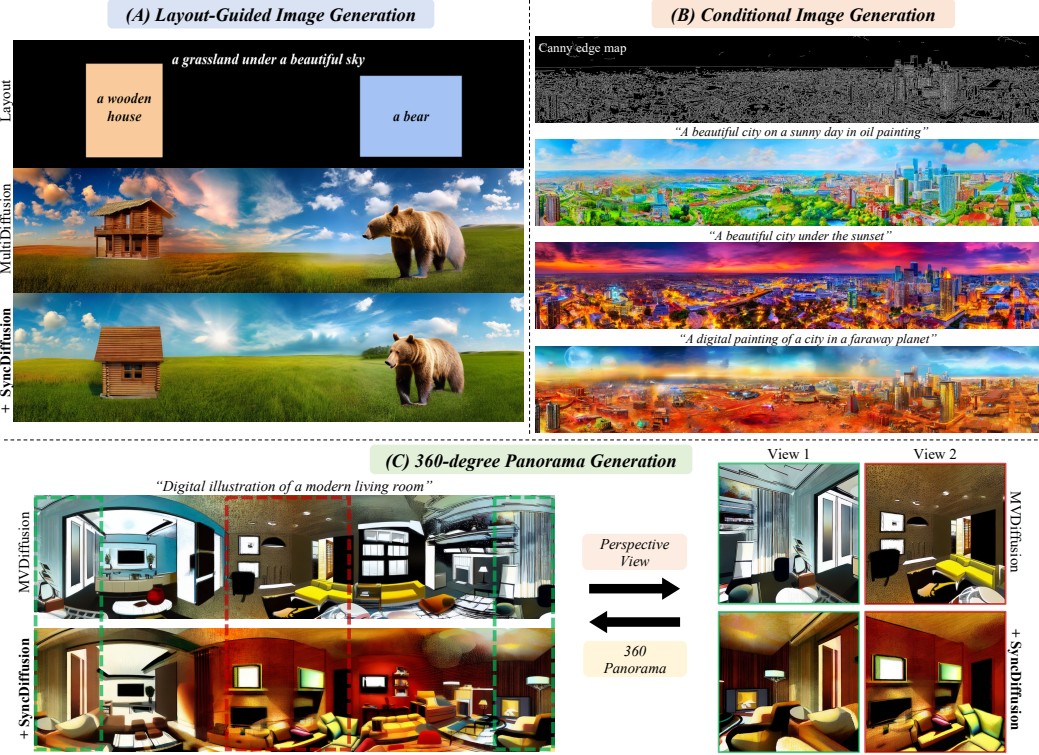

Figure 6: Plug-and-play applications of SYNCDIFFUSION.

from different angles. While View 1 and View 2 from the vanilla MVDiffusion (top row) seem to be from two different rooms, with our method the generated images better depict two views from the same room (bottom row).

**Limitations**   While our SYNCDIFFUSION module can significantly enhance the coherence of generated panoramas, it relies on appropriate input prompts to achieve realistic results, as illustrated in Fig. 7. Also, the SYNCDIFFUSION module that includes a forward pass through the neural network and gradient descent computation introduces additional computational overhead.

**Supplementary**   Due to space constraints, we present the following additional results in the Supplementary: more qualitative comparisons with various prompts (S.1, S.8), details about the quantitative evaluation (S.2), evaluation on different resolutions (S.3), results with Style Loss [14] as the perceptual loss (S.4), an ablation study using Eq. 13 instead of Eq. 14 (S.5), an analysis of computation time (S.6), and details on the user study (S.7).

| *"A red sports car"* | *"A fancy hotel room"* |

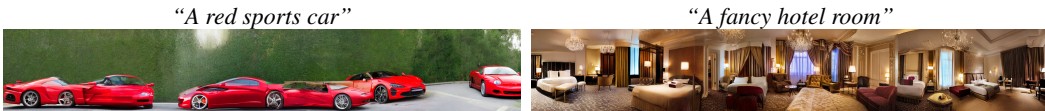

Figure 7: Our failure cases. A suitable input prompt is required to generate realistic panoramas.

# 6   Conclusion

We presented SYNCDIFFUSION, a diffusion synchronization module designed to generate coherent montages through joint diffusions. Using a pretrained diffusion model, we propose guiding the reverse process by updating the noisy images at each intermediate step using gradient descent. This update is based on a perceptual similarity loss calculated with the predictions of the denoised images. Moreover, the idea of SYNCDIFFUSION can be applied to generating textures for 3D models. We plan to investigate such possibilities in future work.

**Potential Negative Societal Impacts**   Image generative models can potentially generate deepfakes, images resembling copyrighted material, biased or discriminatory images, and harmful outputs. Future research is needed to advance the detection of manipulated content and establish societal barriers to protect intellectual property.

## Acknowledgments and Disclosure of Funding

We thank Juil Koo for valuable discussions on diffusion models and Eunji Hong for help in conducting user studies. This work was partially supported by the NRF grant (RS2023-00209723) and IITP grants (2019-0-00075, 2022-0-00594, RS-2023-00227592) funded by the Korean government (MSIT), the Technology Innovation Program (20016615) funded by the Korean government (MOTIE), grants from ETRI, KT, NCSOFT, and Samsung Electronics, and computing resource support from KISTI.

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
