# SyncDiffusion: Coherent Montage via Synchronized Joint Diffusions — Supplementary Material

**Yuseung Lee**   **Kunho Kim**   **Hyunjin Kim**   **Minhyuk Sung**
KAIST
{phillip0701,kaist984,rlaguswls98,mhsung}@kaist.ac.kr

In this supplementary document, we first show more qualitative comparisons with various prompts in Sec. S.1. Sec. S.2 includes a detailed quantitative evaluation of our method with different gradient descent weights ($w = 0, 5, 10, 15$, and $20$). Sec. S.3 shows quantitative evaluation of our method on generating panoramas of different resolutions. In Sec. S.4, we show the comparisons of our method with different perceptual similarity loss functions. Sec. S.5 shows an ablation study result substituting Eq. 14 in the main paper with Eq. 13. Sec. S.6 analyzes the computation time of SYNCDIFFUSION. Sec. S.7 explains the details of our user study. Lastly, Sec. S.8 provides additional qualitative comparisons.

## S.1   More Qualitative Results with Various Prompts

More qualitative results with various prompts are shown in the figures below. The resolutions of images are $512 \times 3072$ for horizontal panoramas and $2048 \times 512$ for vertical panoramas.

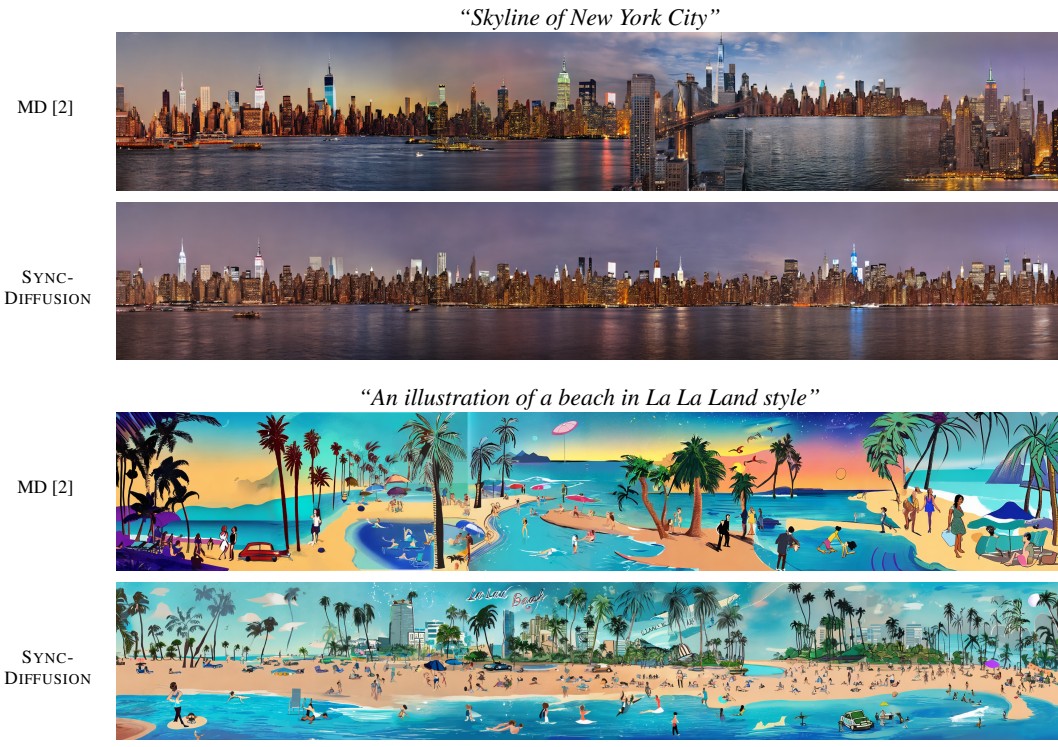

*"Skyline of New York City"*

MD [2]

SYNC-DIFFUSION

*"An illustration of a beach in La La Land style"*

MD [2]

SYNC-DIFFUSION

37th Conference on Neural Information Processing Systems (NeurIPS 2023).

*"A waterfall"*

MD [2]  SYNCDIFFUSION

*"A top view of a single railway"*

MD [2]  SYNCDIFFUSION

*"A photo of a rock concert"*

MD [2]

SYNC-
DIFFUSION

*"Silhouette wallpaper of a dreamy scene with shooting stars"*

MD [2]

SYNC-
DIFFUSION

*"A photo of vines on a brick wall"*                    *"A bird's eye view of an alley with shops"*

MD [2]          SYNCDIFFUSION                           MD [2]          SYNCDIFFUSION

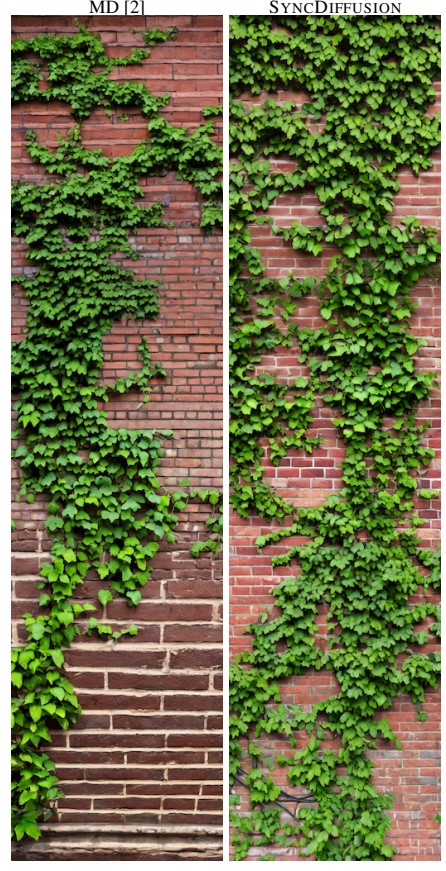

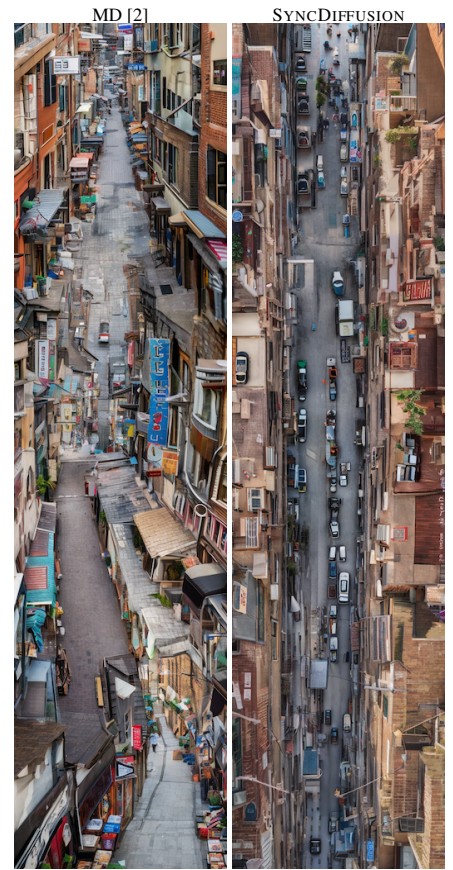

*"A beach with palm trees"*

MD [2]
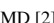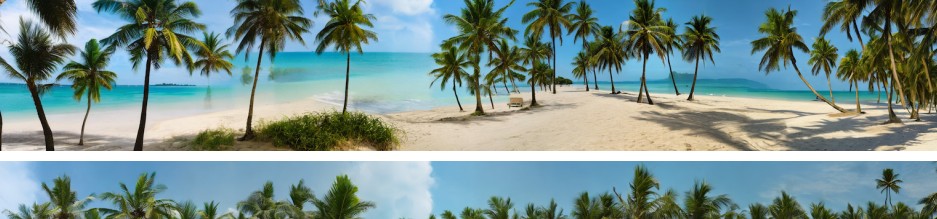

SYNC-
DIFFUSION
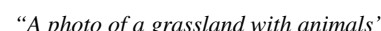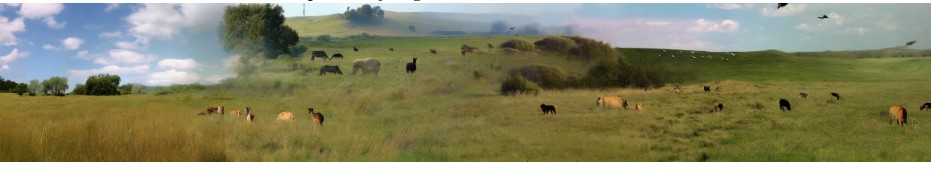

*"A photo of a grassland with animals"*

MD [2]

SYNC-
DIFFUSION
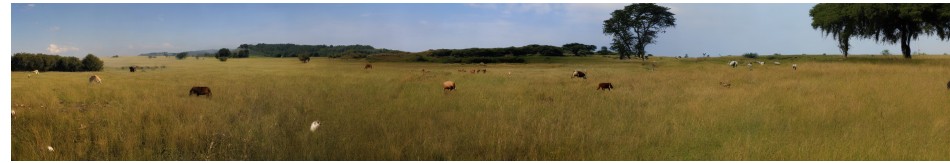

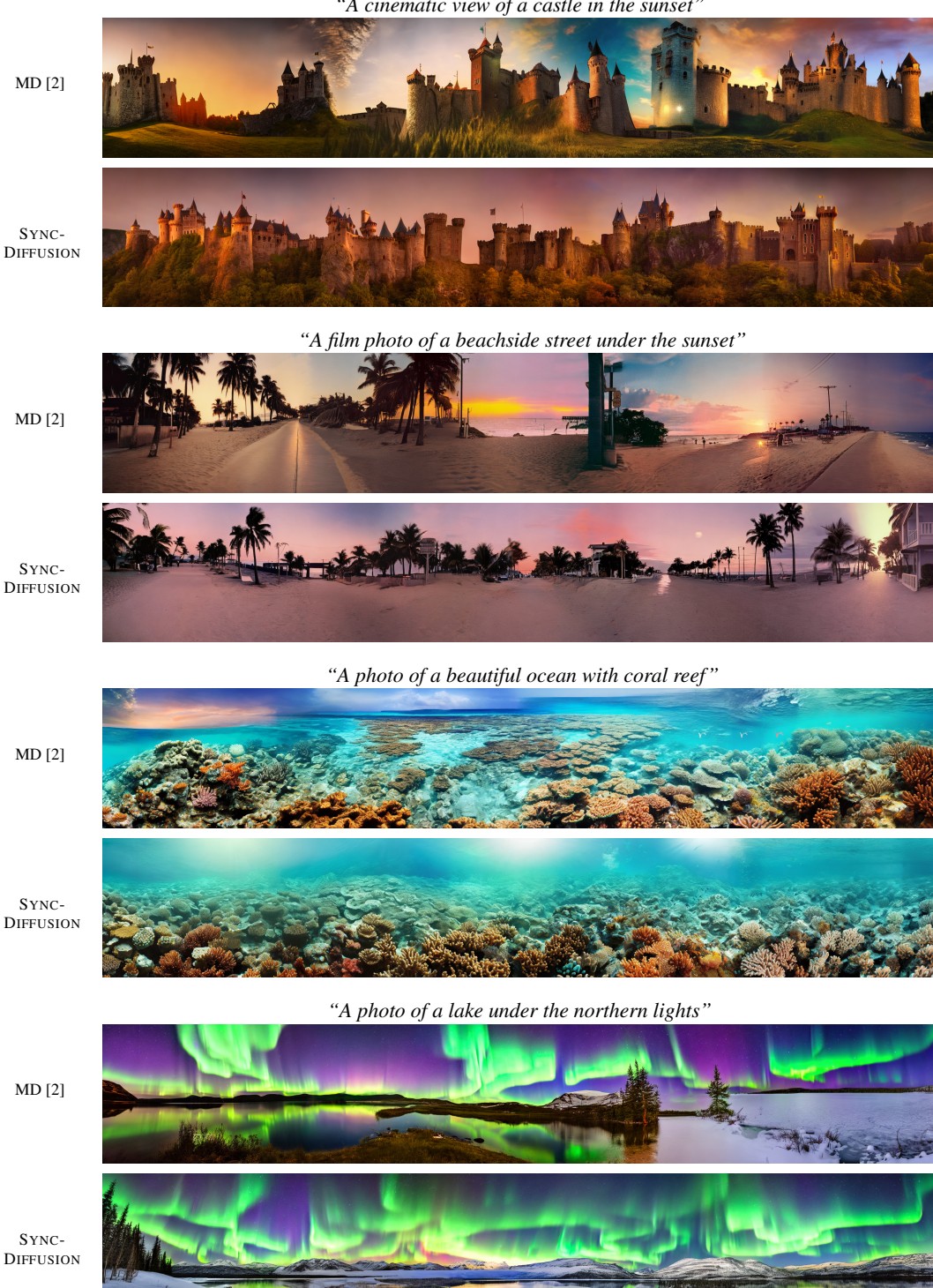

## S.2 Details About Quantitative Evaluation

Tab. S2 shows the detailed quantitative results of SYNCDIFFUSION on panorama generation, reported in Fig. 5 of the main paper. Here we additionally show the results with the gradient descent weight $w = 5$ and $w = 15$, along with the weights $w = 10$ and $w = 20$ reported in Sec. 5 of the main paper. Note that we used KNN-GIQA [4] with $K = 8$ to measure Mean-GIQA in all our experiments. As shown in Tab. S2 (rows 3-7), as the gradient descent weight $w$ increases from 0 to 20, the results of our method display a significant improvement in global coherence, as shown in Intra-LPIPS [6] which decreases from 0.69 ($w = 0$) to 0.56 ($w = 20$), and Intra-Style-L [3] which decreases from 2.98 ($w = 0$) to 1.39 ($w = 20$). These results are more apparent in the line plot of Intra-LPIPS and Intra-Style-L displayed in Fig. S2. Fig. S1 shows the qualitative comparison of the panorama images generated with different weights.

*"A photo of a city skyline at night"*

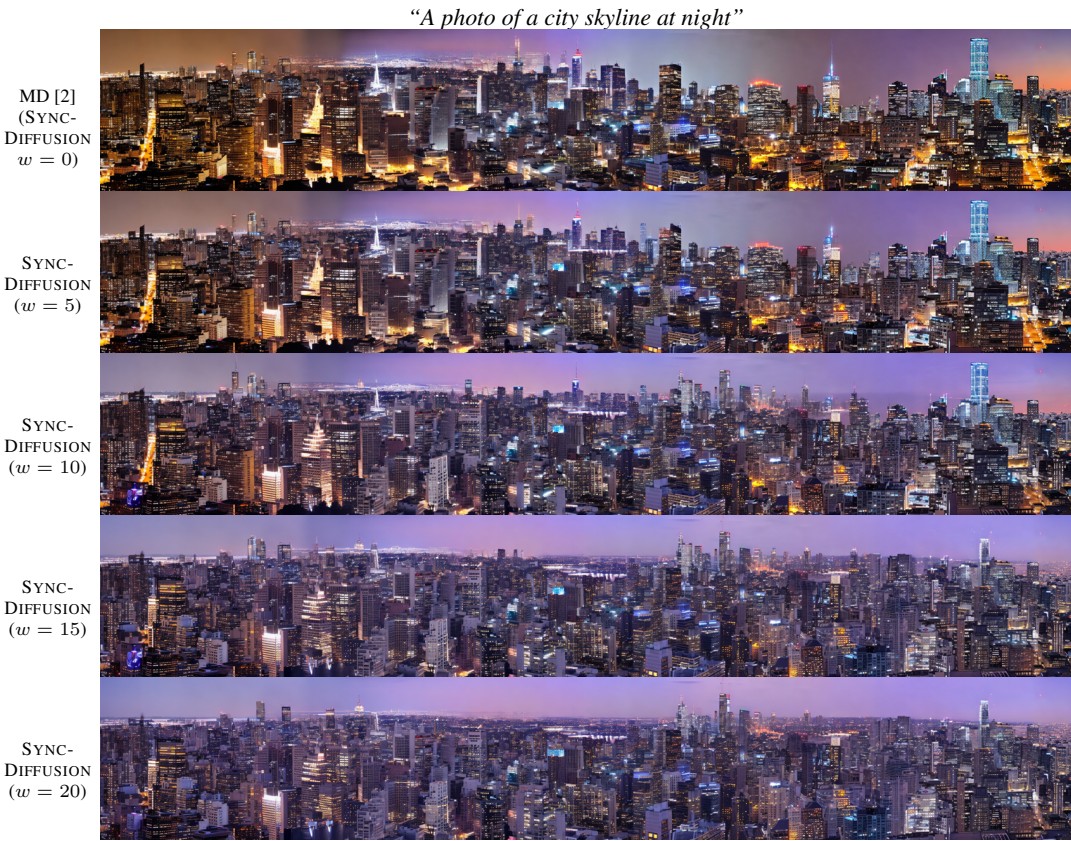

Figure S1: Qualitative comparison of different weights $w$. As $w$ increases, the generated panorama image gradually becomes globally coherent. Compared to MultiDiffusion, as $w$ increases, the left and right sides of the panorama image become more coherent.

## S.3 Quantitative Evaluation on Different Resolutions

We show the quantitative results on different resolutions in Tab. S2 (row 10-13). In addition to the original $512 \times 3072$ resolution, Tab. S2 shows the quantitative comparison of SYNCDIFFUSION and MultiDiffusion [2] for smaller resolution panoramas ($512 \times 2048$ and $512 \times 1024$). In Fig. S2, when comparing the rows 10 and 11, 8 and 9, 3 and 7 respectively, the gap of Intra-LPIPS between our method and MultiDiffusion is preserved (0.13, 0.14, and 0.13, respectively), meaning that our method constantly produces more coherent panoramas than MultiDiffusion regardless of the resolution. The gap of Intra-Style-L between our method and MultiDiffusion even increases as the resolution increases (1.48, 1.57, and 1.59, respectively). On the other hand, the gap of FID and KID between the two methods also increases as the resolution increases: 9.69, 10.26, 11.08 for FID and 7.96, 10.19, 11.96 for KID. We hypothesize that the increase in FID and KID of our method with longer panoramas is due to the tendency that for certain images it is more difficult to find other images that can be merged into a single coherent panorama. The above results indicate that while our method can

| | | Intra-LPIPS ↓ | Intra-Style-L ↓ ($\times 10^{-3}$) | Mean-GIQA ↑ ($\times 10^{-3}$) | FID ↓ | KID ↓ ($\times 10^{-3}$) | Mean-CLIP-S ↑ |
|---|---|---|---|---|---|---|---|
| 1 | SD [5] | $0.74 \pm 0.07$ | $8.40 \pm 6.27$ | $26.70 \pm 6.90$ | $28.31 \pm 10.89$ | $< 0.01 \pm 0.13$ | $31.63 \pm 1.89$ |
| 2 | BLD [1] | $0.58 \pm 0.06$ | $4.64 \pm 3.32$ | $24.27 \pm 6.19$ | $84.29 \pm 36.74$ | $66.54 \pm 37.30$ | $31.41 \pm 1.66$ |
| | | | SYNCDIFFUSION with Various Gradient Descent Weight $w$ (Eq. 14) | | | | |
| 3 | $w = 0$ (MD [2]) | $0.69 \pm 0.09$ | $2.98 \pm 2.41$ | $28.54 \pm 7.99$ | $\mathbf{33.52} \pm \mathbf{12.43}$ | $\mathbf{9.04} \pm \mathbf{4.23}$ | $31.77 \pm 2.32$ |
| 4 | $w = 5$ | $0.64 \pm 0.07$ | $2.15 \pm 1.61$ | $\mathbf{28.58} \pm \mathbf{7.84}$ | $35.57 \pm 12.43$ | $12.09 \pm 4.98$ | $31.85 \pm 2.33$ |
| 5 | $w = 10$ | $0.60 \pm 0.07$ | $1.75 \pm 1.31$ | $28.28 \pm 7.54$ | $38.24 \pm 15.24$ | $15.08 \pm 6.77$ | $\mathbf{31.90} \pm \mathbf{2.33}$ |
| 6 | $w = 15$ | $0.58 \pm 0.06$ | $1.54 \pm 1.21$ | $27.74 \pm 7.19$ | $41.04 \pm 16.74$ | $17.47 \pm 8.29$ | $31.86 \pm 2.25$ |
| 7 | $w = 20$ | $\mathbf{0.56} \pm \mathbf{0.06}$ | $1.39 \pm 1.15$ | $27.17 \pm 6.66$ | $44.60 \pm 18.45$ | $21.00 \pm 11.06$ | $31.84 \pm 2.19$ |
| | | | Panorama Size: $512 \times 2048$ | | | | |
| 8 | MD [2] | $0.69 \pm 0.09$ | $2.96 \pm 2.41$ | $28.33 \pm 7.79$ | $33.07 \pm 12.38$ | $8.58 \pm 3.99$ | $31.77 \pm 2.14$ |
| 9 | SYNCDIFFUSION | $0.55 \pm 0.06$ | $1.39 \pm 1.19$ | $27.08 \pm 6.65$ | $43.33 \pm 17.98$ | $18.77 \pm 10.19$ | $31.77 \pm 2.14$ |
| | | | Panorama Size: $512 \times 1024$ | | | | |
| 10 | MD [2] | $0.66 \pm 0.09$ | $2.57 \pm 1.97$ | $28.17 \pm 7.54$ | $30.66 \pm 11.79$ | $5.24 \pm 3.04$ | $31.73 \pm 2.22$ |
| 11 | SYNCDIFFUSION | $0.53 \pm 0.06$ | $1.09 \pm 0.77$ | $26.41 \pm 6.38$ | $40.35 \pm 16.43$ | $13.20 \pm 7.61$ | $31.71 \pm 2.01$ |
| | | | SYNCDIFFUSION Ablation Study | | | | |
| 12 | Eq. 14 → Eq. 13 | $0.68 \pm 0.09$ | $2.95 \pm 2.39$ | $28.53 \pm 7.99$ | $33.58 \pm 0.09$ | $9.15 \pm 4.25$ | $31.78 \pm 2.32$ |
| 13 | Style Loss [3] | $0.64 \pm 0.10$ | $\mathbf{1.08} \pm \mathbf{1.10}$ | $25.74 \pm 6.31$ | $73.05 \pm 37.56$ | $56.64 \pm 39.58$ | $31.15 \pm 2.32$ |

Table S2: Quantitative results on panorama generation.

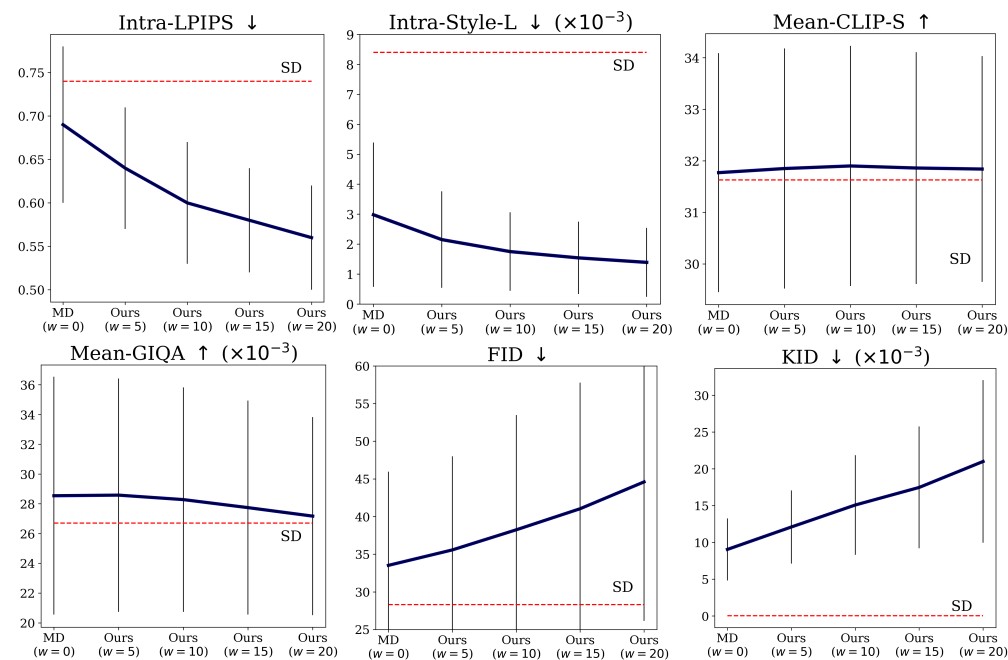

Figure S2: Line plots of the quantitative results shown in Tab. S2 with varying gradient descent weight $w$. The dashed lines (SD) represent the evaluation results of the Stable Diffusion [5] reference set images. The vertical lines represent the standard deviation.

guide the joint diffusion process to generate highly coherent images regardless of the resolution, generating longer panoramas that are globally coherent can lead to a decrease in the diversity of generations, thus resulting in a negative effect on FID and KID.

## S.4 Results of SYNCDIFFUSION with Style Loss

As described in Sec. 4 in the main paper, any off-the-shelf perceptual similarity loss can be utilized in our method. Here we show the results of our method with Style Loss [3] as the loss function $\mathcal{L}$ in Eq. 14 in the main paper. Fig. S3 shows panorama images generated by MultiDiffusion [2], and

our method with LPIPS [6] and Style Loss [3] as the perceptual similarity loss function, respectively. To observe visible changes in the panorama outputs, we multiplied $10^6$ to the Style Loss and set the gradient descent weight $w$ to $0.1$. Tab. S2 (row 13) demonstrates that SYNCDIFFUSION with Style Loss achieves better coherence compared to MultiDiffusion as measured by Intra-LPIPS and Intra-Style-L, while showing a negative effect on the metrics regarding fidelity: Mean-GIQA, FID, and KID. Note that Intra-Style-L is significantly decreased as the guidance was provided with Style Loss. The second row in Fig. S3 shows that Style Loss can guide the joint diffusion processes to generate a globally coherent panorama image, as compared to the MultiDiffusion output in the first row.

*"Natural landscape in anime style illustration"*

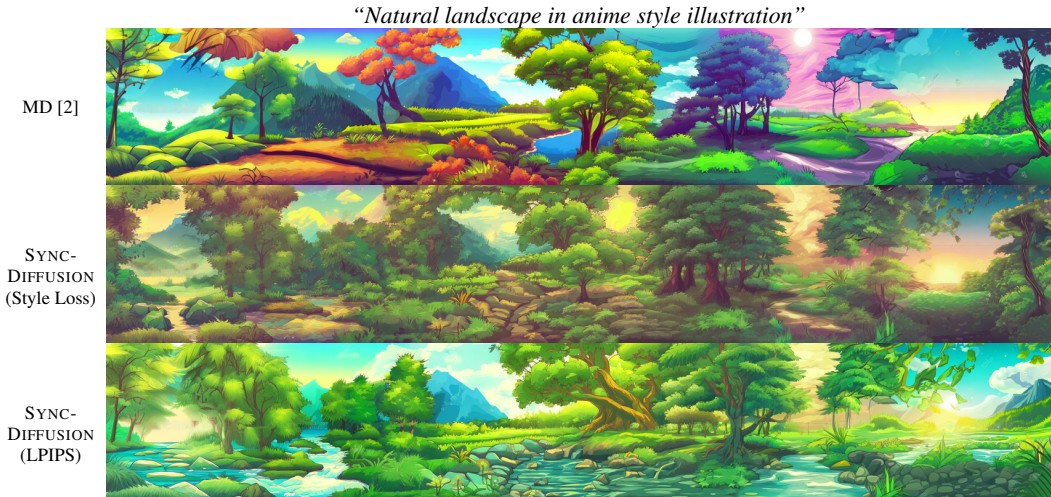

MD [2]

SYNC-
DIFFUSION
(Style Loss)

SYNC-
DIFFUSION
(LPIPS)

Figure S3: Qualitative comparisons of MultiDiffusion and SYNCDIFFUSION with Style Loss.

## S.5 Ablation on Predicting the Foreseen Denoised Observation

Tab. S2 (row 12) shows the quantitative comparison of the panorama generations using our method and after substituting the original Eq. 14 in the main paper with Eq. 13 in which the noisy image $\mathbf{x}_t^{(i)}$ is decoded instead of utilizing the foreseen denoised observation $\phi_\theta(\mathbf{x}_t^{(i)}, t)$. Although Intra-LPIPS is still slightly reduced compared to MultiDiffusion when using Eq. 13, the change is negligible compared to that of the original formulation Eq. 14. This result is straightforward as measuring the perceptual loss between noisy images would not provide meaningful guidance to the diffusion process, whereas comparing the perceptual similarity of *foreseen denoised* observations can give a meaningful guidance for global coherence.

## S.6 Analysis on the Computation Time

As our SYNCDIFFUSION module requires the gradient descent computation, it introduces additional computational overhead during the sampling process. Since our method is based on the DDIM reverse process with 50 timesteps, the gradient descent is applied 50 times. Here we examine two methods to accelerate the generation process while still ensuring a notable improvement in coherence: applying SYNCDIFFUSION on a fixed interval and on the initial sampling steps.

**Fixed interval**  We define $f$ as the frequency of the gradient descent during the DDIM reverse process of SYNCDIFFUSION, with the default value of $f = 50$. Tab. S3 shows the quantitative results and the computation time when the gradient descent is performed 10 times ($f = 10$) and 5 times ($f = 5$) in total with uniform intervals, with the gradient descent weight fixed to $w = 20$. Although applying the gradient descent for every step leads to the highest global coherence with Intra-LPIPS of 0.56 and Intra-Style-L of 1.39, in practice applying the gradient descent for 5 or 10 times can still achieve meaningful improvement in the coherence compared to MultiDiffusion as shown in rows 3-5 of Tab. S3, while reducing the computation time compared to the $f = 50$ case. Note that Intra-LPIPS decreases from 0.69 to 0.62 and Intra-Style-L decreases from 2.98 to 2.14 for $f = 10$.

**Initial steps**  We further analyze the effectiveness of performing the gradient descent for the initial sampling steps. Rows 6-7 of Tab. S3 show the quantitative results and the computation time when the

gradient descent is applied for the initial five and three steps out of the total 50 steps, respectively. The gradient descent weight is fixed to $w = 20$. Comparing row 4 and row 7 shows that by computing the SYNCDIFFUSION function for just the initial three steps is analogous to computing it for ten times at regular intervals in terms of coherence (Intra-LPIPS, Intra-Style-L) and superior in terms of fidelity and diversity (FID and KID), while taking less than 70% of the latter's computation time. The qualitative comparisons of the early-stage synchronization are shown in Fig. S5.

| | Intra-LPIPS ↓ | Intra-Style-L ↓ ($\times 10^{-3}$) | Mean-GIQA ↑ ($\times 10^{-3}$) | FID ↓ | KID ↓ ($\times 10^{-3}$) | Mean-CLIP-S ↑ | Time(s) |
|---|---|---|---|---|---|---|---|
| SD [5] | $0.74 \pm 0.07$ | $8.40 \pm 6.27$ | $26.70 \pm 6.90$ | $\mathbf{28.31} \pm \mathbf{10.89}$ | $\mathbf{<0.01} \pm \mathbf{0.13}$ | $31.63 \pm 1.89$ | - |
| MD [2] | $0.69 \pm 0.09$ | $2.98 \pm 2.41$ | $\mathbf{28.54} \pm \mathbf{7.99}$ | $33.52 \pm 12.43$ | $9.04 \pm 4.23$ | $31.77 \pm 2.32$ | $46.10 \pm 1.07$ |
| SYNCDIFFUSION | | | | | | | |
| $f = 50$ | $\mathbf{0.56} \pm \mathbf{0.06}$ | $\mathbf{1.39} \pm \mathbf{1.15}$ | $27.17 \pm 6.66$ | $44.60 \pm 18.45$ | $21.00 \pm 11.06$ | $31.84 \pm 2.19$ | $339.53 \pm 2.85$ |
| $f = 10$ | $0.62 \pm 0.07$ | $2.14 \pm 1.72$ | $28.43 \pm 7.75$ | $36.22 \pm 14.03$ | $12.84 \pm 5.59$ | $\mathbf{31.85} \pm \mathbf{2.27}$ | $104.83 \pm 3.38$ |
| $f = 5$ | $0.64 \pm 0.07$ | $2.33 \pm 1.83$ | $28.44 \pm 7.85$ | $35.18 \pm 13.31$ | $11.43 \pm 4.68$ | $31.81 \pm 2.24$ | $81.17 \pm 0.53$ |
| Init. 5 Steps | $0.61 \pm 0.06$ | $1.96 \pm 1.36$ | $28.21 \pm 7.48$ | $36.31 \pm 13.83$ | $12.09 \pm 4.76$ | $31.77 \pm 2.25$ | $79.12 \pm 1.72$ |
| Init. 3 Steps | $0.62 \pm 0.06$ | $2.07 \pm 1.40$ | $28.43 \pm 8.19$ | $35.40 \pm 12.99$ | $11.15 \pm 3.76$ | $31.79 \pm 2.26$ | $71.56 \pm 2.64$ |

Table S3: Analysis on the computation time of our SYNCDIFFUSION and MultiDiffusion [2].

## S.7 Details of User Study

For each user study, the order of the images was shuffled. Given a total of 200 questions with a random pair of panoramas, we collected 20 responses each from the participants on Amazon Mechanical Turk who passed our five vigilance tasks. The vigilance tasks were designed to distinguish our outputs from concatenations of Stable Diffusion images generated without joint diffusion. Out of the 100 participants, 86, 90, 84 participants successfully completed all the vigilance tasks for the user study for coherence, image quality and prompt compatibility, respectively.

Fig. S4 shows screenshots of our user study. We set all participants to be Amazon Mechanical Turk Masters who are located in the US. The average time that participants spent on solving a set of 25 problems (including the vigilance tasks) was 248.21 seconds, and we compensated them with a payment of 0.76$ per person. This is equal to 11.02$ per hour, which exceeds the US federal minimum wage.

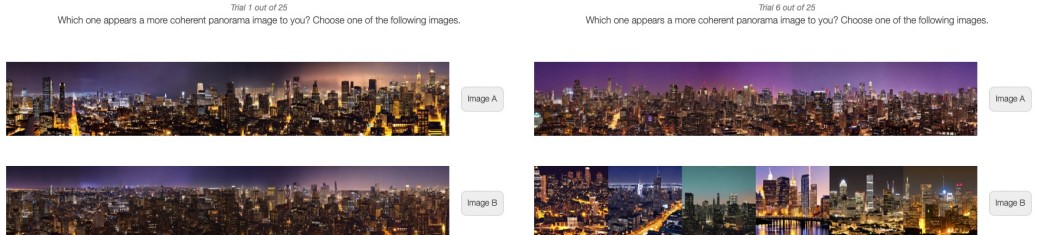

Figure S4: User study screenshots.

*"Natural landscape in anime style illustration"*

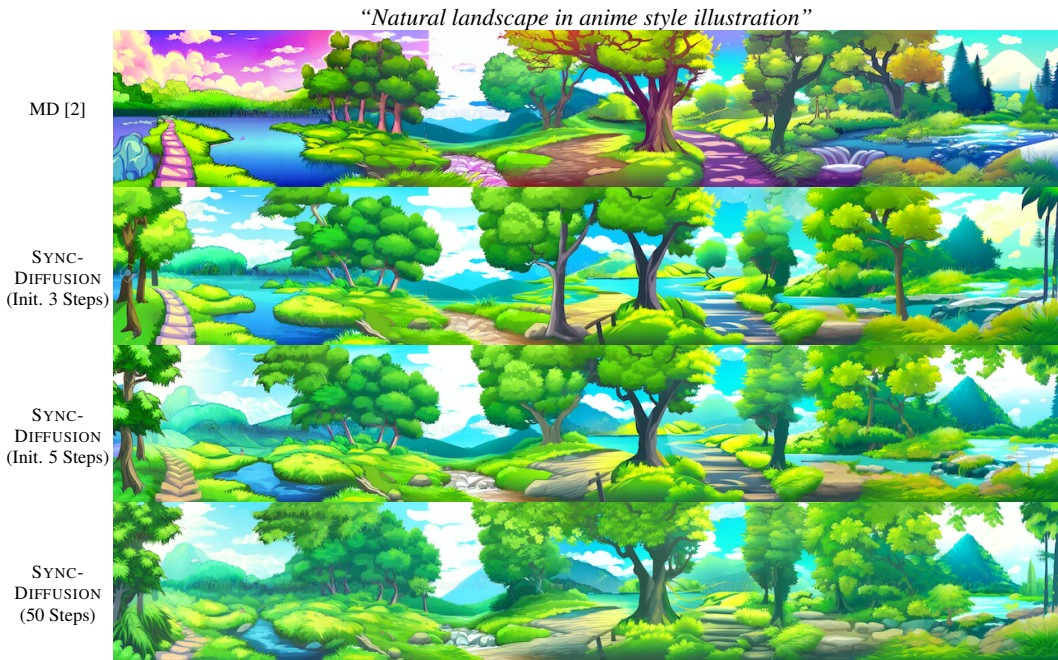

MD [2]

SYNC-DIFFUSION (Init. 3 Steps)

SYNC-DIFFUSION (Init. 5 Steps)

SYNC-DIFFUSION (50 Steps)

Figure S5: Qualitative comparisons of the early-stage synchronization of SYNCDIFFUSION.

## S.8 More Qualitative Results

More qualitative results are shown in the figures below.

| *"A photo of a city skyline at night"* |
| :---: |

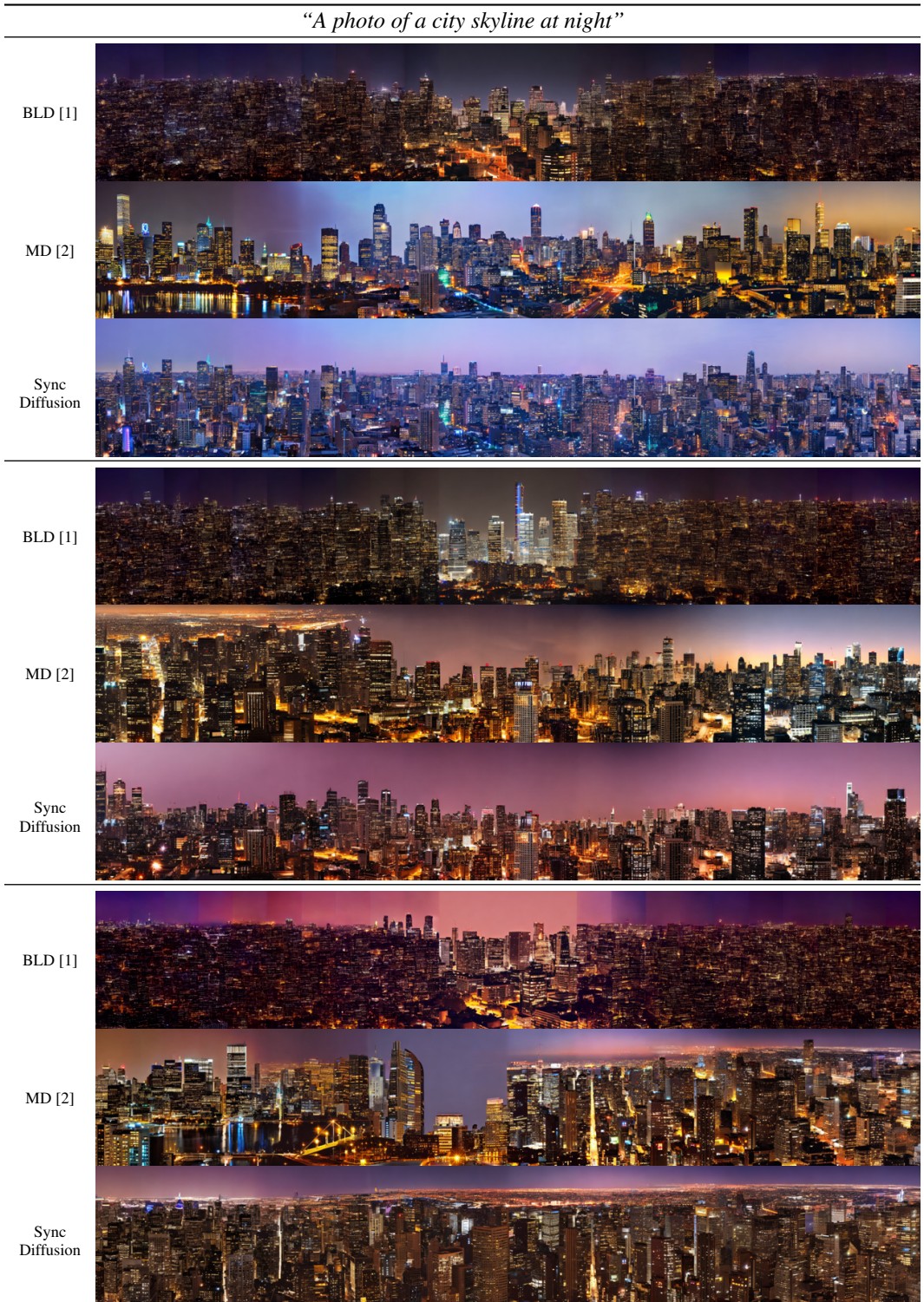

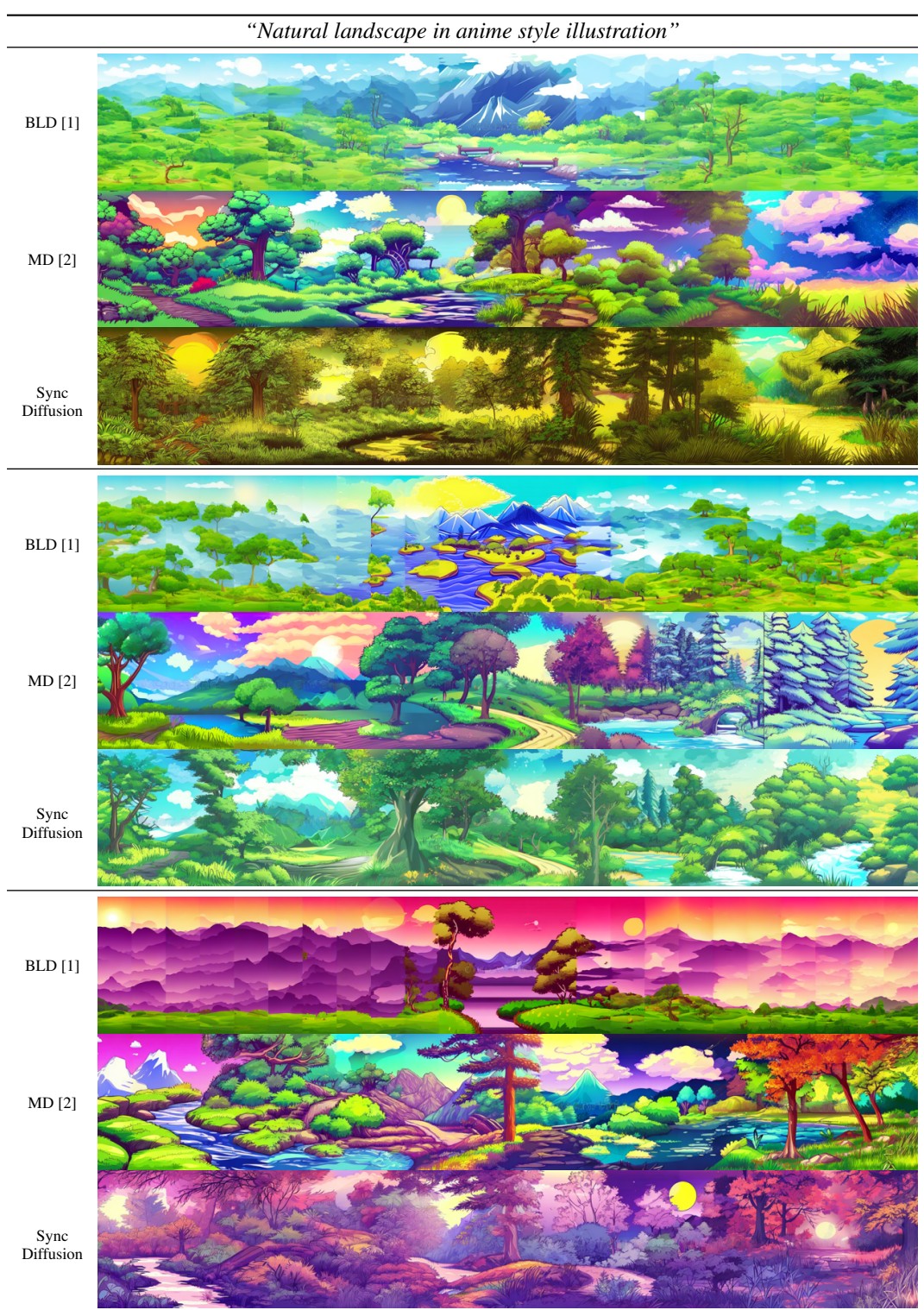

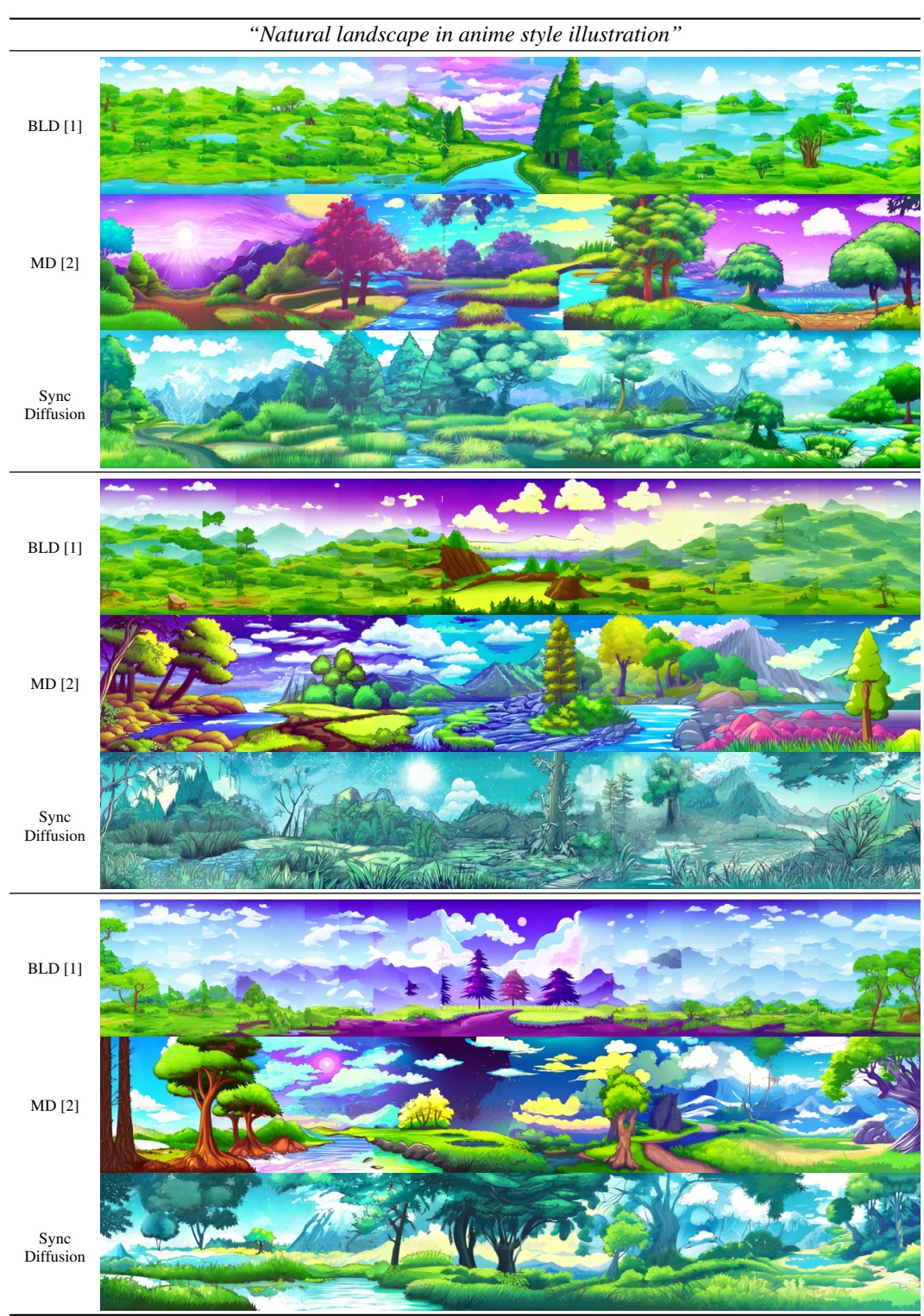

BLD [1]

MD [2]

Sync
Diffusion

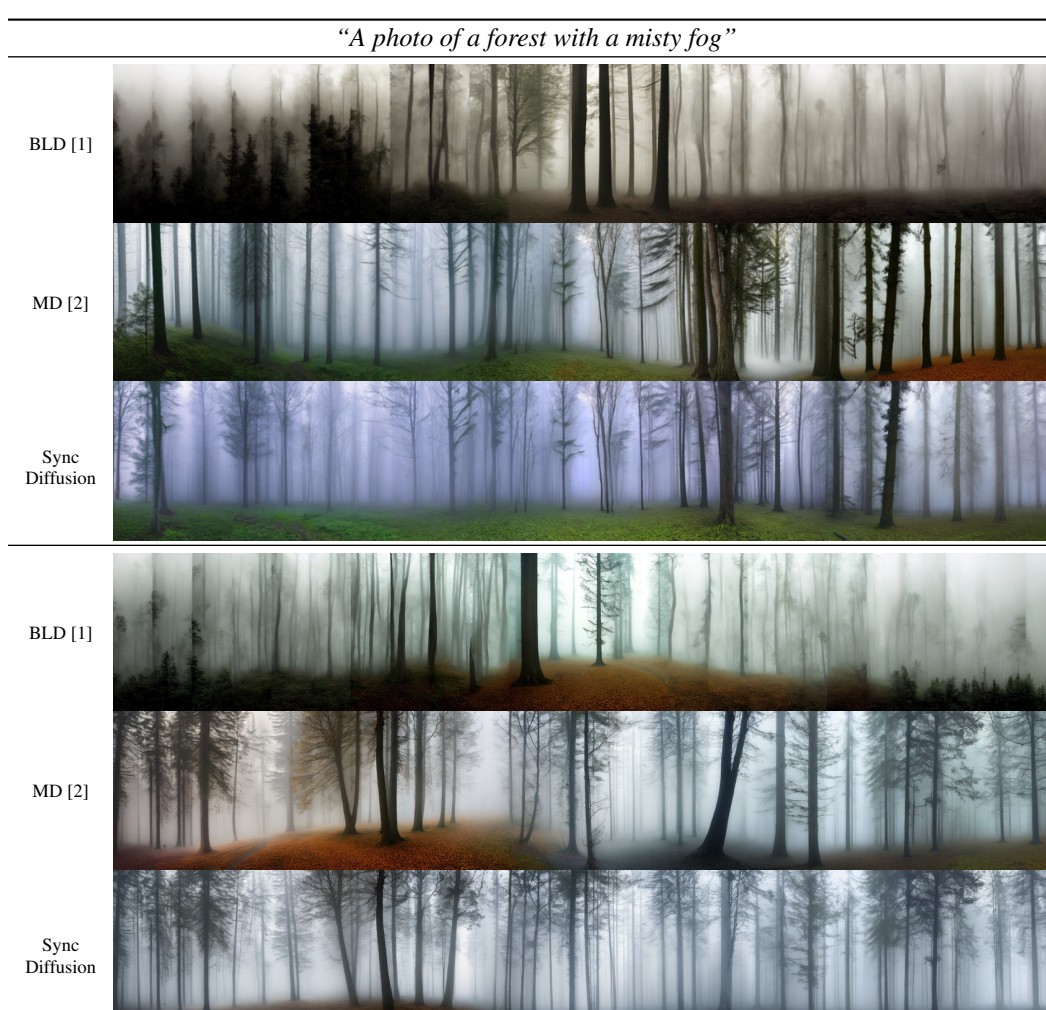

BLD [1]

MD [2]

Sync
Diffusion

*"A photo of a snowy mountain peak with skiers"*

BLD [1]

MD [2]

Sync
Diffusion

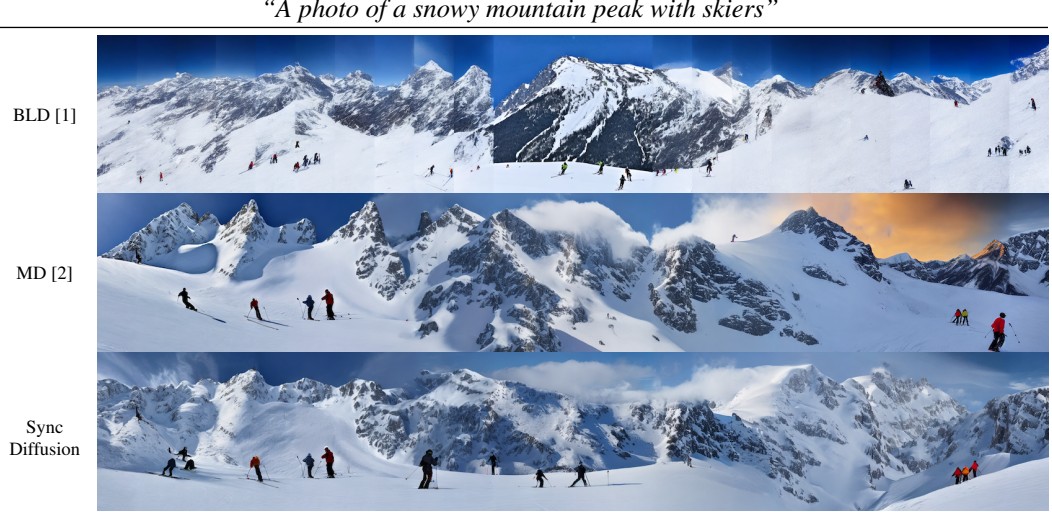

*"A photo of a mountain range at twilight"*

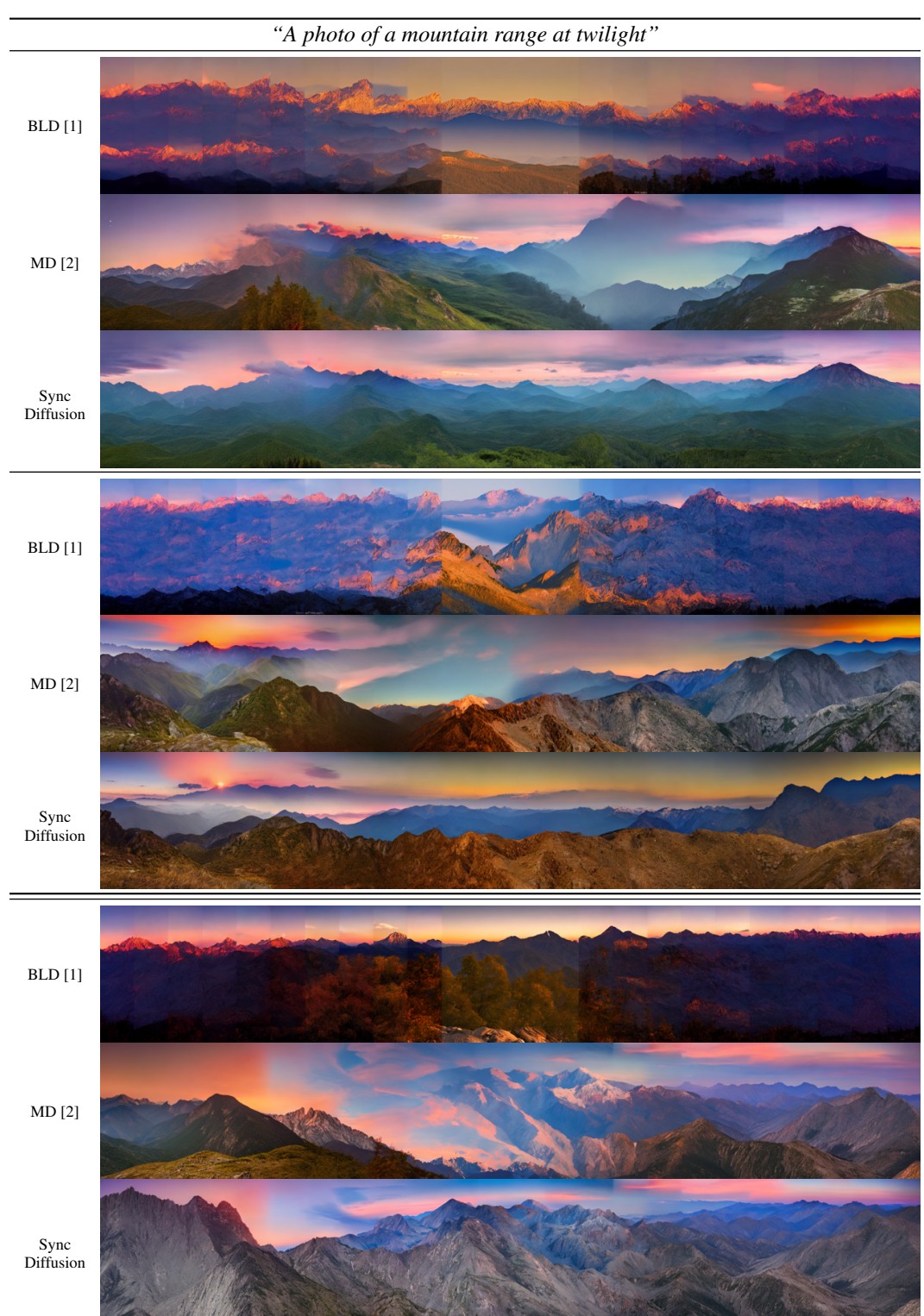

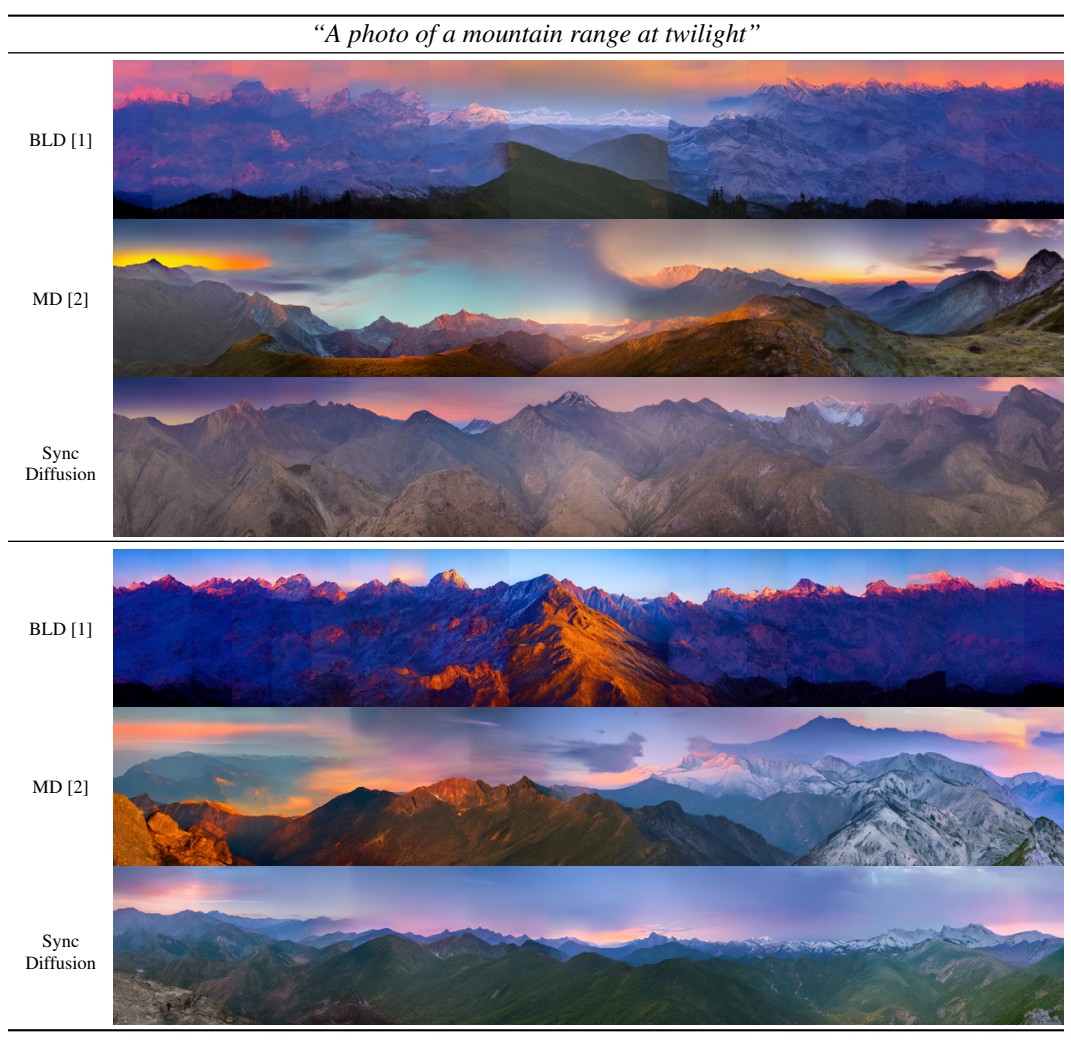

*"Cartoon panorama of spring summer beautiful nature"*

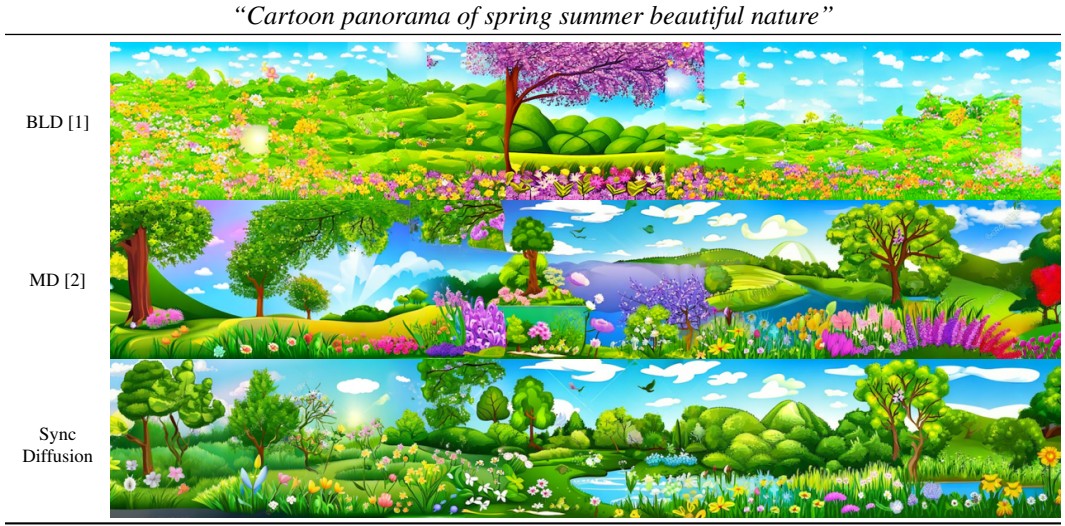