# OpenReview forum: "SyncDiffusion: Coherent Montage via Synchronized Joint Diffusions"
_NeurIPS.cc/2023/Conference — NeurIPS 2023 poster_

### Official Review · Reviewer_DATV · 2023-06-30

**Soundness:** 2 fair
**Presentation:** 3 good
**Contribution:** 2 fair
**Rating:** 6
**Confidence:** 4

**Summary:**

In this paper, the authors propose a new module called SyncDiffusion that enables to synchronize multiple outputs from diffusion models to create coherent montage. This module guides the reverse diffusion process in each window by shifting x_t with gradients of a perceptual loss between the target window and an anchor window. The experimental results show that SyncDiffusion contributes to produce more coherent montage compared to previous methods.

**Strengths:**

- This study adresses an important problem, namely how to synchronize multiple estimates from diffusion models to produce a single output. In particular, how to guarantee a global conherence in the output image is highly non-trivial, but should be critical for practical applications such as panorama image generation.
- The main idea, which is to use gradients of a perceptual loss between target and anchor windows to ensure the global coherence, is simple and can be jointly used with various methods for combining multiple outputs from diffusion models.

**Weaknesses:**

- The current design of the proposed algorithm relies heavily on the homogeneity of the target scene, which may limit its applicability. This is due to LPIPS used for guidance, which forces every window share a similar composition of the image. Personally, I wonder this causes the failure cases shown in Fig. 6.
- The evaluation by Intra-LPIPS and Intra-Style-L is interesting to see a global coherence in a resultant image, but the superiority of the values is not clear, because it is unknown how much the values will be in real panorama images.
- Somewhat related to above, the effectiveness of the proposed method is not clear. In the quantitative comparison, Mean-CLIP-S and Mean-GIQA are on par, and FID is even worse than that of MD (I understood why it happans, though). On the other hand, the user study is conducted only for evaluating the coherence, not for image quality nor prompt fidelity.
- Although the authors argue that SyncDiffusion is a plug-and-play module, it is evaluated only with MultiDiffusion in the experiments.


**Questions:**

- Please see Weaknesses.
- FID of MultiDiffusion in Fig. 5 seems much different from that shown in the original paper [3] (around 30 vs ~10). Why is it?
- It seems straightforward to apply SyncDiffusion to layout-to-image task by just adding an appropriate cropping process before computing the perceptual loss. This should make quantitative comparison much easier as done in MultiDiffusion or [R1]. Why did the authors focus on generating panorama images in the experiments?
  - [R1] "Reduce, Reuse, Recycle: Compositional Generation with Energy-Based Diffusion Models and MCMC," ICLR 2023.

---

Given the authors' response, I updated the rating accrodingly.

**Limitations:**

- Several limitation are described in the manuscript.

---

> ### Author Rebuttal · Authors · 2023-08-09
>
> We greatly appreciate your review, acknowledging that we "address an important problem" which is also "highly non-trivial", and also recognizing that our solution is "simple and can be jointly used with various methods". We address the concerns and questions raised in the review.
>
>
> **(1) Relying on homogeneity limiting its applicability?**
>
> First of all, using LPIPS as the guidance for the synchronization does not result in the failure cases shown in Figure 6. Please refer to Figure 2 in the PDF file attached to the above global response (Additional Experimental Results). You can find the link at the bottom of the text. The figure displays the results of performing MultiDiffusion [1] with the same prompts. These results exhibit the same failure cases due to the description of an "object" instead of a "scene."
>
> Furthermore, the LPIPS-based guidance enforces perceptual similarity across the windows while still allowing for varying content. For instance, the last row in Figure 4 illustrates examples of compositing images with distinct contents such as trees, mountains, flowers, ponds, and the sky in various locations.
>
> We would also like to highlight that our approach is versatile, as it allows for the incorporation of any guidance loss function. As demonstrated in Section S.4 of our supplementary, we have also experimented with Style Loss [2]. Furthermore, the weight assigned to the guidance can be adjusted, as explored in Section S.2 of the supplementary. We intend to continue investigating various types of guidance losses within our SyncDiffusion framework.
>
>
> **(2) Comparison with real panorama images**
>
> Please refer to Section (4) of our global response above (Additional Experimental Results), where we report the quantitative comparison of Intra-LPIPS and Intra-Style-L for our output images and real panorama images. As demonstrated in the results, the comparison with real panorama images shows that SyncDiffusion is capable of generating realistic panorama images.
>
>
> **(3) Mean-CLIP-S and Mean-GIQA are on par, and FID is even worse than that of MultiDiffusion.**
>
> We would like to reiterate our analysis of the trade-off between coherence and diversity, as explained in Figure 5, Table S2, and Figure S2 in our supplementary materials. As the weight for perceptual similarity guidance increases, our quantitative evaluations demonstrate an improvement in coherence (lower Intra-LPIPS and Intra-Style-L scores), while fidelity remains unaffected (Mean-CLIP-S remains unchanged), and diversity decreases (higher FID and KID scores). This outcome is expected, given the greater difficulty in generating coherent images compared to producing incoherent ones. Users can adjust the guidance weight to prioritize either coherence or diversity. Furthermore, our additional user study results also highlight the effectiveness of our method (please refer to the following answer and Section (2) of our global response).
>
>
> **(4) Further User Studies Evaluating Image Quality and Prompt Fidelity**
>
> Please refer to Section (2) of our global response above (Additional Experimental Results), where we report the results of supplementary user studies designed to assess the image quality and prompt fidelity. The results indicate that individuals perceive the images generated by SyncDiffusion as having superior image quality and prompt fidelity compared to the baseline images.
>
>
> **(5) Layout-to-image generation and more applications**
>
> In Section (3) of our previous global response (Additional Experimental Results) and Figure 3 of the PDF file attached to the same response (you can locate the link at the bottom of the text), we present an example of layout-to-image generation using SyncDiffusion. We aim to include additional experimental outcomes along with quantitative assessments in the revised version. We appreciate your feedback.
>
>
> **(6) Difference in FID from MultiDiffusion [1]**
>
> The difference in FID originates from different experimental setups, specifically the different numbers of images (500/prompt in ours vs. 2000/prompt in MultiDiffusion), different resolutions of panorama images (512x3072 in ours vs. 512x4608 in MultiDiffusion), and different strides for the windows in the latent space (16 in ours vs. 8 in MultiDiffusion). We adjusted the experiment setup due to limited computation resources. Moreover, as MultiDiffusion employed eight prompts in their experiments, we utilized six of them since the authors publicly disclosed only these six prompts.
>
> Below, we present the results using a single prompt ("a photo of a mountain range at twilight") when we align our experiment setup with that of MultiDiffusion. The quantitative outcome employing the MultiDiffusion experiment setup still highlights that our SyncDiffusion can significantly reduce Intra-LPIPS while causing only marginal changes to FID. Based on the observations in Section (1) of our global response, we computed the gradient descent for only the initial 5 steps for SyncDiffusion.
>
> | |FID ($\downarrow$)|Intra-LPIPS ($\downarrow$)|
> |:--|:--:|:---:|
> |Stable Diffusion| 5.09 | 0.728 |
> |MultiDiffusion| 11.04 | 0.640 |
> |SyncDiffusion (Ours)| 13.04 | 0.547 |
>
> **References**
>
> [1] Bar-Tal, Omer, et al. "Multidiffusion: Fusing diffusion paths for controlled image generation." ICML (2023).
>
> [2] Gatys, Leon A., Alexander S. Ecker, and Matthias Bethge. "Image style transfer using convolutional neural networks." CVPR (2016).

---

> > ### Comment · Reviewer_DATV · 2023-08-12
> > **Reply**
> >
> > Thanks for the authors' response. I have read it as well as other reviews.
> >
> > I would like to thank the authors for providing the additional experimental results. Specifically, given the user study on image quality and quantitative comparison with real panorama images, the effectiveness of the proposed method becomes much clear. It is still a bit less clear whether it is really a good idea to use LPIPS for the guidance, but exploring various guidance loss for better conherence would be one of the future work as the authors stated in the rebuttal.
> >
> > I will update my rating accordingly. Thanks.

---

### Official Review · Reviewer_SgT1 · 2023-07-03

**Soundness:** 3 good
**Presentation:** 3 good
**Contribution:** 3 good
**Rating:** 6
**Confidence:** 4

**Summary:**

This paper proposes a diffusion synchronization module, a plug-and-play module for text-to-panorama generation. The key idea is based on the observation that predicted denoised images closely resemble the final outputs even at the beginning of the denoising process. Authors proposed to synchronize multiple diffusions through gradient descent from a perceptual similarity loss. Experiments demonstrate the effectiveness of the proposed method in generating coherent montages with good fidelity and compatibility with prompts.

**Strengths:**

(1)	This work is well motivated based on the observation that Predicted denoised images closely resemble the final outputs even at the beginning of the denoising process. In addition, it also helps local seamlessness and global coherence.
(2)	Evaluation metrics used in quantitative comparison experiments are elaborate.
(3)	The paper has high writing quality which makes it easy to understand.


**Weaknesses:**

(1)	Based on MultiDiffusion, the author introduces guidance information provided solely by gradient descent from a perceptual similarity loss. I wonder if this guidance is too weak or have they explored other guidance?
(2)	Pretrained stable diffusion v2 is used to generate 512x512-sized images. How can stable diffusion be utilized to generate wider images? Please provide a detailed explanation in the paper. The setting of text-to-panorama generation task should be further clarified.
(3)	Through the quantitative comparisons in Fig. 5, it can be observed that the setting of gradient descent weights has significant impact on the generation results. However, it is unclear how this affects the qualitative or visual effects. I wonder if the authors have experimented with other weights and is it possible that a weight other than w=20 could yield better results.
Additional Comments
(1)	L171, L179, writing errors, Fig. 4 -> Fig. 2
(2)	L191, L198, writing errors, Fig. 4 -> Fig. 3


**Questions:**

See weakness.

**Limitations:**

Yes, authors mentioned the limitation of the proposed in its high sensitivity to prompt and additional computation overhead.
They also mentioned its potential negative sociatal impact on deepfake.

---

> ### Author Rebuttal · Authors · 2023-08-09
>
> We appreciate your review, recognizing that "evaluation metrics are comprehensive" and "the paper is of high writing quality." We address specific queries below.
>
> **(1) Is the guidance not strong enough?**
>
> We have quantitatively and qualitatively showcased the impact of our synchronization guidance in Figures 1, 4, and 5 in the main paper, as well as in the supplementary results. We would like to draw your attention to our extensive qualitative comparison in Section S.1 of our supplementary, where we illustrate the significant distinction between MultiDiffusion [1] and our SyncDiffusion across a variety of examples.
>
> **(2) Other guidance**
>
> In Section S.4 of our supplementary, we presented results obtained by employing our SyncDiffusion method using Style Loss [2] instead of LPIPS. The qualitative and quantitative outcomes are depicted in Figure S3 and Table S2, respectively. These results illustrate that this alternative guidance can also enhance global coherence in the output panorama images, albeit at the expense of greater trade-offs in terms of fidelity and diversity (FID and KID) compared to when LPIPS is used in SyncDiffusion. We plan to further investigate the potential of alternative guidance losses.
>
> **(3) Details about wide image generation**
>
> Here we briefly clarify our algorithm outlined in Sections 3 and 4. For a given target resolution of a panorama image—512x3072 in our experiments—we define a set of windows spanning the image, each being of Stable Diffusion's resolution, i.e., 512x512, and mutually overlapping. In our experiments, this set of windows is established with a stride of 16 in the latent space. We execute the reverse process of Stable Diffusion within each window. However, preceding each step of this reverse process, as detailed in Algorithm 1, we apply our SyncDiffusion module, computing gradient descent with the loss defined across these windows. Furthermore, after each step of the reverse process, we apply the idea of MultiDiffusion [1] and average the latent features of images in the regions of overlap.
>
> **(4) Impacts of the weight ‘w’**
>
> The impact of the weight 'w' is analyzed in Section 5 of the paper and also in Section S.2 of our supplementary. Qualitative and quantitative experimental results obtained by varying the weight 'w' are presented in Figure S1, Table S2, and Figure S2. As expected, the plots in Figure 5 and Figure S2 (which include more variations of ‘w’) in the supplementary illustrate a discernible trend: as the weight increases, global coherence improves (Intra-LPIPS and Intra-Style-L decrease), while fidelity and diversity experience more compromise (FID and KID increase). We anticipate observing a similar trend as the weight 'w' is further increased.
>
> **References**
>
> [1] Bar-Tal, Omer, et al. "Multidiffusion: Fusing diffusion paths for controlled image generation." ICML (2023).
>
> [2] Gatys, Leon A., Alexander S. Ecker, and Matthias Bethge. "Image style transfer using convolutional neural networks." CVPR (2016).

---

> > ### Comment · Reviewer_SgT1 · 2023-08-14
> > **Response to rebuttal**
> >
> > I quite appreciate authors could further clarify their method and address most of my concerns. Therefore I lean to upgrade my score.

---

### Official Review · Reviewer_AhXa · 2023-07-06

**Soundness:** 3 good
**Presentation:** 3 good
**Contribution:** 3 good
**Rating:** 7
**Confidence:** 3

**Summary:**

The paper tackles the problem of image diffusion models for generation montages/panoramas. Their proposed method, dubbed SyncDiggusion, improves on the state-of-the-art (both qualitatively, through user studies, and in quantitative metrics) by synchronizing multiple diffusions through gradient descent using a perceptual similarity loss. This provides a guidance in the reverse diffusion process by adjusting intermediate noisy images at each step. Notably, off-the-shelf perceptual similarity losses can be interchangeably used with this method.

**Strengths:**

1. The paper proposes a novel and interesting solution to the problem of consistent panoramic image generation using diffusion models
1. The method is original, to my knowledge
1. The paper is well written and clear
1. The paper presents adequate theoretical and algorithmic explanations
1. The paper presents a comprehensive set of both qualitative and quantitative results that demonstrate the benefits of their method over previous state-of-the-art, highlighting their contributions well

**Weaknesses:**

1. Coherence is measured not with the current noisy color images $\{D(x_t^{(i)})\}$ but with the predicted denoised color images $\{D(\phi(x_t^{(i)}, t))\}$ (L195). What is the computational overhead of this computation? Similarly, What is the added computational overhead of this method as compared to the reported baselines? A discussion of this is omitted, although mentioned in brief in the limitations sections.
1. In turn, a discussion of the tradeoffs between consistency vs. diversity vs. computational overhead would then be beneficial.

**Questions:**

1. Can this work be extended to other applications such as videos?
1. It was mentioned that choice of prompt is critical in this method, and can easily break it. How were the prompts for the user study chosen? Choice of prompts would greatly influence the outcome of the study, potentially in favour of the proposed algorithm.


**Limitations:**

Yes, limitations are discussed in brief.

---

> ### Author Rebuttal · Authors · 2023-08-09
>
> We sincerely appreciate your review, recognizing the novelty and originality of our method, as well as the comprehensive qualitative and quantitative results. Here, we address the concerns and questions that have been raised.
>
> **(1) Tradeoffs between consistency vs. diversity vs. computational overhead**
>
> We draw your attention to Section S.6 in our supplementary material, where we analyze the trade-offs among coherence, diversity, and computation overhead by adjusting the frequency of gradient descent computation in the reverse process. The results reveal that reducing the frequency of gradient descent computation can lower computation overhead and also impact fidelity and diversity (Mean-GIQA, FID, and KID), while still enhancing coherence (Intra-LPIPS, Intra-Style-L). Notably, performing gradient descent not at every timestep, but only ten times out of 50 timesteps, can decrease Intra-LPIPS from 0.69 to 0.62, while utilizing fewer than 1/3 of the computation time compared to gradient descent at every timestep.
>
> Moreover, in Section (1) of our global response above (Additional Experimental Results), we present the results when involving the computation of the SyncDiffusion function only at the initial 3 or 5 steps of the reverse process. The results demonstrate that the early-stage synchronization is even more effective in improving coherence. Specifically, the initial 3-step synchronization yields better results than a 10-step synchronization with uniform intervals. This finding further contributes to the reduction of computational overhead.
>
> **(2) Application to video generation**
>
> While we haven't yet attempted to apply this concept to video generation, we believe that it holds potential for enhancing the video generation process as well. We appreciate your comment and will delve deeper into this direction for further exploration.
>
> **(3) Choice of prompts**
>
> In Section 5, for the purpose of comparisons, we employed the same prompts that were utilized in MultiDiffusion [1], aiming to eliminate any potential biases in the comparisons. Note that we utilized only six out of the eight prompts that were originally used in MultiDiffusion, as the authors did not provide the remaining two prompts.
>
> **References**
>
> [1] Bar-Tal, Omer, et al. "Multidiffusion: Fusing diffusion paths for controlled image generation." ICML (2023).

---

> > ### Comment · Reviewer_AhXa · 2023-08-12
> > **Reply**
> >
> > Thank you for the response.
> >
> > The additional results are appreciated and shed further light on the effectiveness of this method, as mentioned by reviewer DATV. I have additionally read the responses to the other reviewers. Thank you for the clarification on the user study prompts, I suspected as much. My concerns with computational overhead remain however the reviewers do in fact discuss these concerns well and no one paper is expected to solve all problems. I am more confident now in upgrading my rating to a 7.
> >
> > Thanks.

---

### Official Review · Reviewer_X4Nm · 2023-07-07

**Soundness:** 4 excellent
**Presentation:** 4 excellent
**Contribution:** 3 good
**Rating:** 7
**Confidence:** 4

**Summary:**

The paper addresses the problem of lack of global coherence in panoramas generated with diffusion models (in a tiled manner with overlapping regions). This work builds upon MultiDiffusion [1], which proposed the technique of averaging multiple predictions for an overlapping region at each denoising step to avoid edge artifacts. However, MultiDiffusion lacked long range coherence.
This paper proposes interleaving an additional guidance step using perceptual similarity loss to ensure that each region is coherent with an anchor region.

[1] MultiDiffusion: Fusing Diffusion Paths for Controlled Image Generation, Bar-Tal et al, 2023

**Strengths:**

- The solution of guiding the denoised image with a perceptual similarity loss for global coherence is simple yet effective
- Qualitative and quantitative results are thorough and back the effectiveness of the approach in achieve global coherence
- Paper is very well written, adequately discusses background, related work and limitations.

**Weaknesses:**

- This slows down the inference process as the authors discuss in the supplementary

**Questions:**

None

**Limitations:**

Yes

---

> ### Author Rebuttal · Authors · 2023-08-09
>
> We greatly appreciate your review, acknowledging that our method is "simple yet effective", the experimental results support “the effectiveness of the approach," and the "paper is well-written."
>
> Indeed, the computational overhead is a drawback of our method. However, we would like to reiterate the discussion in Section S.6 of our supplementary, where we highlight that computing the gradient descent not at every timestep, but rather at only a few steps with regular intervals, can still enhance coherence (for instance, reducing Intra-LPIPS by 0.07 by performing the gradient descent every 5 timesteps), while alleviating the computational overhead and also minimizing its impact on fidelity and diversity.
>
> Moreover, in Section (1) of our global response above (Additional Experimental Results), we present the results when involving the computation of the SyncDiffusion function only at the initial 3 or 5 steps of the reverse process. The results demonstrate that the early-stage synchronization is even more effective in improving coherence. Specifically, the initial 3-step synchronization yields better results than a 10-step synchronization with uniform intervals. This finding further contributes to the reduction of computational overhead.
>
> We plan to further explore methods for accelerating the synchronization computation through recent distillation training approaches, as presented in "Consistency Model" [1] or "On Distillation of Guided Diffusion Models" [2].
>
> **References**
>
> [1] Song, Yang, et al. "Consistency models." ICML (2023).
>
> [2] Meng, Chenlin, et al. "On distillation of guided diffusion models." CVPR (2023).

---

> > ### Comment · Reviewer_X4Nm · 2023-08-13
> >
> > Thank you for your response and for further clarifying the concern around latency. I am happy to keep my original rating.

---

### Author Rebuttal · Authors · 2023-08-09

We extend our gratitude to all the reviewers for their invaluable feedback. Here, we summarize our additional experimental results.

**[Additional Experiment Results]**

**Please refer to the attached document for the figures.**

**(1) Speeding up with early-stage synchronization**

In addition to the computation time analysis in Section S.6 of the supplementary, where we explored the variation in the frequency of gradient descent computation (SyncDiffusion Function in Algorithm 1) during the reverse process, we conducted further experiments involving the computation of the SyncDiffusion function only at the early stage of the reverse process.

The table below presents the results when the gradient descent computation is performed at each time step for the initial three or five steps, out of the total 50 steps. Comparing row 4 and row 7 shows that by computing the SyncDiffusion function for just the initial three steps is analogous to computing it for ten times at regular intervals in terms of coherence (Intra-LPIPS, Intra-Style-L) and superior in terms of fidelity and diversity (FID and KID), while taking less than 70% of the latter’s computation time. The qualitative comparisons of the early-stage synchronization are shown in Figure 1 of the attached PDF (you can locate the link at the bottom of this global response).

| |Intra-LPIPS ($\downarrow$)|Intra-Style-L ($\downarrow$) ($\times 10^{-3}$)|Mean-GIQA ($\uparrow$) ($\times 10^{-3}$) | FID ($\downarrow$) | KID ($\downarrow$) ($\times 10^{-3}$) | Mean-CLIP-S ($\uparrow$) | Time (s)|
|:--|:--:|:---:|:---:|:---:|:---:|:---:|:---:|
|Stable Diffusion|0.74|8.40|26.70|28.31|<0.01|31.63|-|
|MultiDiffusion|0.69|2.98|**28.54**|**33.52**|**9.04**|31.77|46.10|
|Ours (f=50)|**0.56**|**1.39**|27.17|44.60|21.00|31.84|339.53|
|Ours (f=10)|0.62|2.14|28.43|36.22|12.84|**31.85**|104.83|
|Ours (f=5)|0.64|2.33|28.44|35.18|11.43|31.81|81.17|
|**Ours (Init. 5 Steps)**|0.61|1.96|28.21|36.31|12.09|31.77|79.55|
|**Ours (Init. 3 Steps)**|0.62|2.07|28.43|35.40|11.15|31.79|71.20|


**(2) Further User Studies Evaluating Image Quality and Prompt Fidelity**

The table presented below showcases the results of supplementary user studies designed to assess image quality and prompt fidelity. To evaluate image quality and prompt fidelity, we employed the same user study setup utilized in our previous experiment, while changing the questions to the ones used in Imagen [2] for image quality and DALL-E [3] for prompt fidelity:

* Image Quality: "Which panorama image is of higher quality? Choose one of the following images."
* Prompt Fidelity: "Which panorama image best matches the shared caption? Choose one of the following images."

The results affirm that human evaluators perceive SyncDiffusion results as demonstrating superior image quality and higher prompt fidelity compared to the MultiDiffusion outcomes.

| |Coherence (%)|Image Quality (%)|Prompt Fidelity (%)|
|:--|:--:|:---:|:---:|
|MultiDiffusion| 33.65 | 42.81 | 40.50|
|SyncDiffusion (Ours)| **66.35** | **57.19** | **59.50**|

**(3) Layout-to-image and more applications**

In Figure 3 of the attached PDF below (you can locate the link at the bottom of this global response), we showcase a result of layout-to-image generation and offer a qualitative comparison with MultiDiffusion. Note that MultiDiffusion generates an unnatural image with an incoherent background, whereas our SyncDiffusion produces a superior outcome with a coherent background. In the revised version, we aim to incorporate more experimental results for layout-to-image generation along with quantitative evaluations.

**(4) Comparison with real panorama images**

The table below demonstrates a comparison of Intra-LPIPS and Intra-Style-L for our output images and the real panorama images. Since there is no existing benchmark for real panorama images (especially non-360 flat panorama images), we collected 100 images from Unsplash with a width at least twice larger than the height. Then, we scaled the images to match the height of our generated panorama images. For a fair comparison, we also randomly selected 100 generated panorama images and then cropped them along the width to align the resolution with that of the corresponding real images.

As demonstrated in the results below, our SyncDiffusion achieves similar or even slightly lower Intra-LPIPS and Intra-Style-L scores compared to the real images. In contrast, MultiDiffusion yields much higher Intra-LPIPS and Intra-Style-L scores. This illustrates that our SyncDiffusion is capable of generating more realistic panorama images.

| |Intra-LPIPS ($\downarrow$)|Intra-Style-L ($\downarrow$) ($\times 10^{-3}$)|
|:--|:--:|:---:|
|Stable Diffusion| 0.74 | 8.4 |
|MultiDiffusion| 0.67 | 2.6 |
|SyncDiffusion (Ours)| 0.55 | 1.1 |
|Real Panorama| 0.51 | 1.6 |

**References**

[1] Bar-Tal, Omer, et al. "Multidiffusion: Fusing diffusion paths for controlled image generation." ICML (2023).

[2] Saharia, Chitwan, et al. "Photorealistic text-to-image diffusion models with deep language understanding." NeurIPS (2022).

[3] Ramesh, Aditya, et al. "Zero-shot text-to-image generation." ICML (2021).

---

### Decision · Program_Chairs · 2023-09-21

**Decision:**

Accept (poster)

**Comment:**

The AC has carefully read the paper, reviews, author responses, and discussions. This paper tackles the problem of image diffusion models for generation montages/panoramas. SyncDiggusion improves on the state-of-the-art (both qualitatively, through user studies, and in quantitative metrics) by synchronizing multiple diffusions through gradient descent using a perceptual similarity loss. All reviewers are positive towards acceptance. The AC agrees with the reviewers that this is a solid submission to NeurIPS and thus recommends acceptance with a certain confidence.